



# Application and evaluation of the dendroclimatic process-based model MAIDEN during the last century in Canada and Europe

Jeanne Rezsöhazy[1,2], Hugues Goosse[1], Joël Guiot[2], Fabio Gennaretti[3], Etienne Boucher[4], Frédéric André[5], and Mathieu Jonard[5]

[1]Université catholique de Louvain (UCLouvain), Earth and Life Institute (ELI), Georges Lemaître Centre for Earth and Climate Research (TECLIM), Place Louis Pasteur, B-1348 Louvain-la-Neuve, Belgium
[2]Aix Marseille University, CNRS, IRD, INRA, College de France, CEREGE, Aix-en-Provence, France
[3]Institut de recherche sur les forêts, UQAT, Rouyn-Noranda, Québec, J9X 5E4, Canada
[4]Université du Québec à Montreal, Dépt. of Geography and GEOTOP, Montreal, H2V 1C7, Canada
[5]Université catholique de Louvain (UCLouvain), Earth and Life Institute (ELI), Croix du Sud 2, L7.05.09, B-1348 Louvain-la-Neuve, Belgium

**Correspondence:** J. Rezsöhazy (jeanne.rezsohazy@uclouvain.be)

**Abstract.** Tree-ring archives are one of the main sources of information to reconstruct climate variations over the last millennium with annual resolution. The links between tree-ring proxies and climate have usually been estimated using statistical approaches, assuming linear and stationary relationships. Both assumptions may be inadequate but this issue can be overcome by ecophysiological modelling based on mechanistic understanding. In this respect, the model MAIDEN (Modeling and Analysis In DENdroecology) simulating tree ring growth from daily temperature and precipitation, considering carbon assimilation and allocation in forest stands, may constitute a valuable tool. However, the lack of local meteorological data and the limited characterisation of tree species traits can complicate the calibration and validation of such complex model, which may hamper paleoclimate applications. The goal of this study is to test the applicability of the MAIDEN model in a paleoclimate context using as a test case tree ring observations covering the twentieth century from twenty-one Eastern Canadian taiga sites and three European sites. More specifically, we investigate the model sensitivity to parameters calibration and to the quality of climatic inputs and evaluate the model performance using a validation procedure. We also examine the added value of using MAIDEN in paleoclimate applications compared to a simpler tree-growth model, VS-Lite. A bayesian calibration of the most sensitive model parameters provides good results at most of the selected sites with high correlations between simulated and observed tree-growth. Although MAIDEN is found to be sensitive to the quality of the climatic inputs, simple bias-correction and downscaling techniques of these data improve significantly the performance of the model. The split-sample validation of MAIDEN gives encouraging results but requires long tree-ring and meteorological series to give robust results. We also highlight a risk of overfitting in the calibration of model parameters that increases with short series. Finally, MAIDEN has shown higher calibration and validation correlations in most cases compared to VS-Lite. Nevertheless, this latter model turns out to be more stable over calibration and validation periods. Our results provide a protocol for the application of MAIDEN to potentially any site with tree-ring width data in the extratropical region.





## 1 Introduction

Instrumental data inform on past climate only back to the nineteenth century because few continuous records exist before this period (Harris et al., 2014; University of East Anglia Climatic Research Unit (CRU), 2017). Complementary, indirect climate

records from natural archives such as tree rings offer a longer-term perspective. In this context, dendroclimatology, defined as the science that allows the inference of past climates from tree-rings, enables climate reconstructions at high temporal resolution (annual), over several centuries or millennia (Fritts, 1976; Hughes et al., 2011). Thanks to the availability of tree-ring observations in many regions, they represent the main data source in most large scale hemispheric reconstructions covering the last millennium (e.g. Cook et al., 1999; Jones et al., 2009; Mann et al., 2009; Anchukaitis et al., 2017; Wilson et al., 2016;

PAGES 2k Consortium, 2017; St. George and Esper, 2019; Esper et al., 2018).

Reconstructing past climate on the basis of tree-rings first requires to establish a relationship between measured variables, such as tree ring width or density, and climate. This has been classically done using statistical approaches (Fritts, 1976; Cook and Kairiukstis, 1990), often reducing the problem to empirical linear relationships. Consequently, numerous temperature reconstructions are based on multiple linear regressions, calibrated using temperature during the instrumental period (e.g.

Fritts, 1991; Jones et al., 1998; Mann et al., 1999, 2008). When using those statistical models for the entire period covered by dendroclimatic data, we assume both linear and stationary relationships (Guiot et al., 2014), while those assumptions may be inadequate for many records (Briffa et al., 1998; Wilson and Elling, 2004; Wilson et al., 2007; D'Arrigo et al., 2008).

Process-based tree-growth models are able to overcome those limitations of statistical models, by explicitly representing the processes at the origin of the recorded signal (Guiot et al., 2014). They are also one kind of a larger group of models

called Proxy System Models (PSM). PSMs simulate the development of measured variables (here, in tree rings) based on climatic variables as inputs. They integrate a simplified representation of the mechanisms governing the relationship between climate and observations used to capture paleoclimatic information (Evans et al., 2013). These models can be applied in an inverse mode to estimate the climatic conditions that gave rise to the measured characteristics (Guiot et al., 2014; Boucher et al., 2014). Alternatively, they can be forced by climate model results (direct mode), allowing thereby to compare model results with

indirect climate records, without the need to reconstruct the climate from these observations (Evans et al., 2013; Dee et al., 2016). In addition to major advantages for model-data comparisons, proxy system models can facilitate the assimilation of proxy data in long climate model runs (Dee et al., 2016; Goosse, 2016). In paleoclimatology, the objective of data assimilation is to optimally combine the results of one climate model and the observations to obtain an estimate of the state of the climate system as accurate as possible (Kalnay, 2003). This technique is now used regularly to obtain reanalysis providing estimates of

different climatic variables, such as temperature, precipitations, atmospheric and ocean circulation for the last decades. Similar procedures are being developed in palaeoclimatology (e.g Goosse et al., 2012; Franke et al., 2017; Tardif et al., 2018) but so far, all tests using actual data have been based on temperature reconstructions derived from proxies, not on proxies themselves. This implies additional uncertainties when reconstructing temperatures.



Several models developed to simulate tree growth have been applied in dendroclimatology (Guiot et al., 2014). Among them,
the VS-Lite model is a deterministic numerical model that simulates the primary response of ring width to climate based on the
principle of limiting climatic factors (i.e. temperature and soil moisture; Tolwinski-Ward et al., 2011). Because of its simplicity
and the small number of inputs required, it has been used in a wide range of conditions in a large number of paleoclimate
studies (e.g Breitenmoser et al., 2014; Lavergne et al., 2015; Dee et al., 2016; Steiger and Smerdon, 2017; Seftigen et al., 2018;
Fang and Li, 2019). However, VS-Lite is not able to reproduce tree-growth observations for numerous sites, particularly when
the dependence on climatic conditions is complex (Breitenmoser et al., 2014). More comprehensive models such as the full
Vaganov-Shashkin model (Vaganov et al., 2006) or MAIDEN (Modeling and Analysis In DENdroecology; Misson, 2004) could
be more adapted to those conditions. One of the strenghts of the MAIDEN model is to include the influence of atmospheric
$CO_2$ concentration on growth. This is essential when we know that the atmospheric concentration of $CO_2$ increased by 30%
during the last fifty years (Myhre et al., 2013; Boucher et al., 2014). Unfortunately, those models including explicitly complex
biological processes such as photosynthesis and carbon allocation may need careful initialisation and calibration for each
set. They may thus require specific information on the sites that may not be available. This may then hamper a systematic
application of the model on a large number of sites as done for instance with VS-Lite (Breitenmoser et al., 2014).

Before applying a mechanistic model to a wide range of tree ring records covering the past centuries, testing its applicability
over the twentieth century when data allow an estimation of the model skill appears necessary, which is the goal of our
study. For a specific study site, local meteorological data and measurements of several ecophysiological variables allow a
precise calibration of many individual processes included in the model. However, this is a rare case and likely one of the main
limitations in the application of the model to a wide range of sites and soil conditions or when driven by climate model results
that have known biases (Flato et al., 2013). We first present in Sect. 2.1 the two dendroclimatic models that are compared in this
study, namely the complex model MAIDEN and the more simple model VS-Lite. MAIDEN and VS-Lite are applied to selected
sites of the Northern Hemisphere (described in Sect. 2.2), covering a range of environmental conditions and tree species. A
first set of data consists of a large number of sites from the same region with similar environmental conditions but with low
in situ replication, while a second set only contains a few sites but with good replication. In this way, we test the applicability
of MAIDEN to two datasets contrasted in terms of site documentation that allow us to evaluate the extent to which MAIDEN
can be applied. We compare the calibration methods adopted for VS-Lite (Tolwinski-Ward et al., 2013) and MAIDEN (Hartig
et al., 2019) in Sect. 2.3. Different strategies to select the value for the most sensitive parameters of the MAIDEN model as
well as the sensitivity of parameters calibration to the quality of climatic inputs are tested in Sect. 3.1, 3.2 and 3.3. Finally, we
compare calibration and validation statistics of both models and discuss their applicability to a wide range of sites and species
in Sect. 3.4 and 3.5.



## 2 Material and Methods

### 2.1 Tree growth models

#### 2.1.1 The MAIDEN model

The dendroclimatic model MAIDEN has initially been developed by Misson (2004). It explicitly includes biological processes, namely photosynthesis and carbon allocation to different tree compartments, to simulate an annual tree growth increment. The model uses daily climatic inputs (i.e. $CO_2$ atmospheric concentration, precipitations and minimum and maximum air temperature). Up to now, MAIDEN has been applied in the Mediterranean (Gea-Izquierdo et al., 2015) and temperate regions (Misson, 2004; Boucher et al., 2014), in the Eastern Canadian taiga (Gennaretti et al., 2017) and in Argentina (Lavergne et al., 2017). Currently, there are two versions of the model from Gea-Izquierdo et al. (2015), developed for the Mediterranean forests, and Gennaretti et al. (2017) for boreal tree species. A unified version from those two versions has also been developed by Fabio Gennaretti (unpublished). In this study, all tests have been conducted using the unified version of MAIDEN. This unified version gives the opportunity to choose between the version from Gennaretti et al. (2017) and from Gea-Izquierdo et al. (2015) and, if needed, to test equations from both versions to evaluate their impact. However, here, only the version from Gennaretti et al. (2017) is used as it is the most adapted to the selected sites.

MAIDEN simulates photosynthesis on a daily basis and allocates the daily available carbon from photosynthesis and stored non-structural carbohydrates to different pools (leaves, roots, stem and storage). The allocation is based on functionnal rules defined following the ongoing phenological phase (five phases per year: winter 1, winter 2, budburst, summer and fall). At the end of the year, the model sums all the daily carbon inputs allocated to the stem to get an annual tree growth increment (yearly Dstem, hereafter Dstem, in grams of carbon per square meter of stand per year). Dstem is assumed to be proportional to tree-ring growth so that we can build simulated tree-ring index time series and compare it with tree-ring width (hereafter TRW) observations (Sect. 2.3.1) (Gea-Izquierdo et al., 2015; Gennaretti et al., 2017).

Tree-ring observations site and climate station (corresponding to a single location or grid cell as a function of the climatic dataset) constants of the MAIDEN model (Table S1) are derived from observations, as far as possible. For practical reasons, we were not able to retrieve slope and aspect informations from a Digital Elevation Model, for example, because it requires field knowledge of the site and for each sample, that we cannot systematically obtain, given our global scale goals. Thus, slope and aspect constants are set to zero. The soil is divided in four layers (1-15cm; 15-30cm; 30-65cm; 65-100cm). Clay and sand fractions are extracted from the Harmonized World Soil Database (hereafter HWSD) v1.2 at 30 arc-second resolution (FAO/IIASA/ISRIC/ISSCAS/JRC, 2012) at the nearest cell with observed value which is always at a distance smaller than 100 km to the site and assigned as follows: 1-30cm parameters from the HWSD for the two first soil layers in MAIDEN; 30-100cm parameters from the HWSD for the two deepest soil layers in MAIDEN. Soil layers thickness is fixed at the same values for all sites, as for fine roots fractions.





### 2.1.2 The VS-Lite model

VS-Lite was developed by Tolwinski-Ward et al. (2011) as a simplified version of the full Vaganov-Shashkin model (Vaganov
et al., 2006). The model reproduces the primary response of ring width to climate using an approach based on the limiting
factors principle (i.e. temperature and soil moisture) and on threshold growth response functions. It does not model any biolog-
ical processes explicitly so that it cannot be considered fully mechanistic. The model needs monthly climate data (cumulated
precipitations and average temperature) as inputs as well as latitude of the study site. The main output of VS-Lite used here is
a unitless annual tree-growth increment (Tolwinski-Ward et al., 2011).

## 2.2 Study sites and climate data

### 2.2.1 Study sites

A network of tree-ring width chronologies of *Picea mariana* collected in similar conditions is available for the Eastern Canadian
taiga (Nicault et al., 2014; Boucher et al., 2017). Those chronologies have been standardized using the Age-Band Regional
Curve Standardization (or RCS) method (Briffa et al., 2001). We also use the Eastern Canadian taiga chronology for *Picea
mariana* from Gennaretti et al. (2017) (hereafter *QC_taiga*), standardized using a site-specific RCS (Gennaretti et al., 2014b).
The latter is highly replicated (Gennaretti et al., 2014b) compared to the other Eastern Canadian sites from Nicault et al. (2014)
and Boucher et al. (2017), which cover a broader spatial range, and provides additional observations to calibrate the model.
From this network, we have only selected sites from Nicault et al. (2014) and Boucher et al. (2017) ending at least in 2000, with
an expressed population signal (defined as the amount of variance of a population chronology infinitely replicated explained
by a finite subsample; Buras, 2017) equal to or above 0.8 and replication equal to or above 15. We have also kept the site
from Gennaretti et al. (2017) as a control site. At the end of the selection process, we get twenty-one sites (Fig. 1a). In order
to increase replication, the Canadian sites from Nicault et al. (2014) and Boucher et al. (2017) are aggregated based on an
one degree grid by averaging tree-ring width chronologies (Fig. 1b). From this, we get five aggregated sites (Table 1). Note
that *QC_taiga* is not included into the aggregation process to keep it as a reference. This observational network represents an
archetypal example of a singular species that covers an important hydroclimatic gradient, which makes it a relevant candidate
for our calibration and validation exercises.

Three additionnal tree-ring width chronologies (hereafter European sites) are used to perform tests on sites with good repli-
cation, especially at the European Alps site, and long nearby series from meteorogical stations (Fig. 2): EALP (47N/10.7E;
2050m; European Alps; *Larix decidua*; Büntgen et al., 2011); SWIT179 (46.77N/9.82E; 1800m; *Picea abies*; standardized
with a cubic-smoothing spline with a 50% frequency cut-off at 35 years; Seftigen et al., 2018) and FINL045 (68.07N/27.2E;
*Pinus sylvestris*; standardized using a spline with a 50% frequency cut-off response at 32 years; Babst et al., 2013). Those three
European sites exemplify a situation where we only have access to individual sites with different species and from different
environmental conditions that are not part of a larger network of tree-ring width observations.





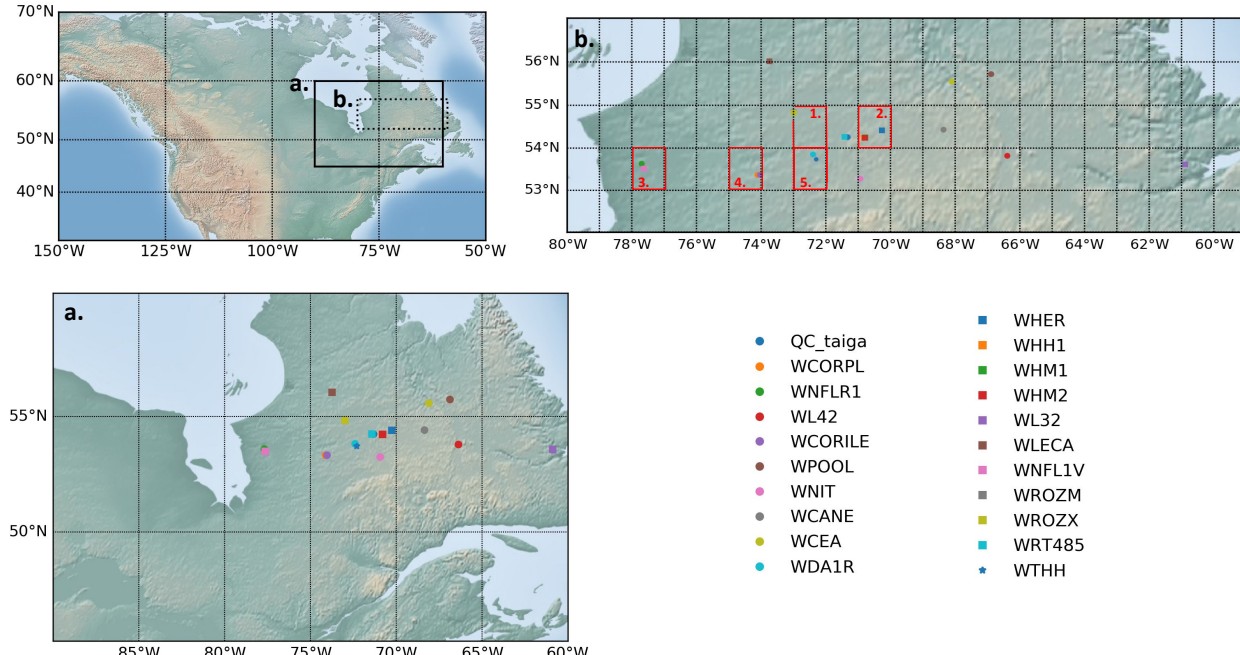

**Figure 1.** Location of (a) twenty-one Eastern Canadian taiga sites (20 sites from Nicault et al. (2014) and Boucher et al. (2017); 1 site called here *QC_taiga* from Gennaretti et al. (2017)) (b) aggregated Eastern Canadian taiga sites from Nicault et al. (2014) and Boucher et al. (2017) based on a 1°grid (red numbered grid cells). Background map from Hunter (2007).

**Table 1.** Aggregated Eastern Canadian taiga sites based on the individual sites from Nicault et al. (2014) and Boucher et al. (2017) (Fig. 1a and b).

| Aggregated site name | Individual sites | Grid cell number |
|---|---|---|
| WROZ | WROZM, WROZX | 1 |
| WH | WHER, WHH1, WHM1, WHM2 | 2 |
| WNFL | WNFL1V, WNFLR1 | 3 |
| WCOR | WCORILE, WCORPL | 4 |
| WDA1R_WTHH | WDA1R, WTHH | 5 |

### 2.2.2 Climate data

Daily climatic inputs are needed to run MAIDEN (Sect. 2.1.1). Monthly climatic inputs for VS-Lite are computed from those daily data. Three daily climatic datasets with different spatial resolution (Table 2) were selected for our analysis on the East-



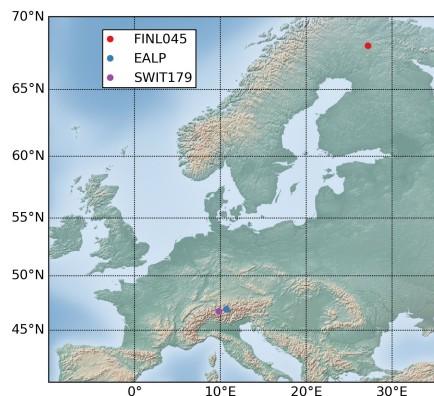

**Figure 2.** Location of three European sites with tree-ring width observations used in this study. Background map from Hunter (2007).

ern Canadian taiga network (Fig. 1a and b). First, a dataset at a high spatial resolution of 5 minutes from the gridded in-
terpolated Canadian database of daily minimum-maximum temperature and precipitation (Hutchinson et al., 2009, hereafter
NRCAN). The *Global Meteorological Forcing Dataset for land surface modeling* (http://hydrology.princeton.edu/data.php;
Sheffield et al., 2006) at 1° resolution is used as a mid-resolution climatic dataset (hereafter GMF). The NOAA-CIRES 20th
Century Reanalysis V2c (https://www.esrl.noaa.gov/psd/data/gridded/data.20thC_ReanV2c.monolevel.html) at 2° resolution
is used as a low-resolution dataset (hereafter 20CRv2c). Finally, the 20CRv2c dataset was modified to match the monthly mean
seasonal cycle of the high-resolution dataset NRCAN (hereafter 20CRv2c corr.). This simple bias correction and downscaling
to the location of the site is done by removing the difference between the monthly mean seasonal cycle of 20CRv2c (2°)
and NRCAN (5') from the maximum and minimum temperature data. In order to avoid negative values, daily precipitations
are multiplied by the ratio between the monthly mean seasonal cycle of NRCAN (5') and 20CRv2c (2°). The time series are
extracted from the grid cells nearest to the studied individual sites. The climatic data are averaged over the individual sites data
for the aggregated Eastern Canadian sites (Table 1).

The Global Historical Climate Network Daily (Table 2; see Table S3 for details on selected stations; Menne et al., 2012a, b;
hereafter GHCN) is used to perform analysis on the European sites (FINL045, EALP, SWIT179, Fig. 2).

Daily atmospheric $CO_2$ concentration data are linearly interpolated from the annual data from Sato and Schmidt (https:
//data.giss.nasa.gov/modelforce/ghgases/).

## 2.3   Calibration

### 2.3.1   The MAIDEN model

We have developed a protocol to systematically and automatically calibrate the model, through a bayesian procedure with
Markov Chain Monte Carlo sampling carried out using the DREAMzs algorithm (Hartig et al., 2019). The calibration procedure
focusses on the most sensitive parameters of the model identified in Gennaretti et al. (2017): six parameters influencing the



**Table 2.** Description of all daily climatic datasets used in this study (Abbreviation, Climatic dataset, Spatial resolution and Source), time periods on which MAIDEN and VS-Lite simulations are performed with each specific climatic dataset (Time period) and sites where climate data are used (Sites). European and Canadian sites refer to Fig. 1 and 2 respectively.

| Abbreviation | Climatic dataset | Spatial resolution | Source | Time period | Sites |
|---|---|---|---|---|---|
| GHCN | Global Historical Climate Network Daily | station | Menne et al., 2012a, b | 1909-1944 or 1910-1949;1950-2000 | European sites |
| NRCAN | Canadian database of daily minimum-maximum temperature and precipitation | 5 minutes | Hutchinson et al., 2009 | 1950-2000 | Canadian sites |
| GMF | Global Meteorological Forcing Dataset for land surface modeling | 1 degree | Sheffield et al., 2006 | 1950-2000 | Canadian sites |
| 20CRv2c | NOAA-CIRES 20th Century Reanalysis V2c | 2 degrees | NOAA-CIRES | 1950-2000;1900-2000 | Canadian sites |
| 20CRv2c corr. | NOAA-CIRES 20th Century Reanalysis V2c corrected for bias in the mean seasonal cycle based on NRCAN | 2 degrees | - | 1950-2000;1900-2000 | Canadian sites |

simulated stand growth primary production and twelve parameters involved in the modelling of the daily quantity of carbon allocated to different tree compartments (Table S2). Those 6+12 parameters are calibrated by comparison between simulated Dstem and tree-ring width observations. The comparison relies on the computation of the model likelihood defined as the sum of the logarithms of the normal probability densities of the residuals between the model simulation and the observations. The prior distributions of the 6+12 parameters are assumed to be uniform over an acceptable range, as in Gennaretti et al. (2017).

The calibration procedure is made up of three steps. During the first step, we calibrate the twelve carbon allocation parameters, while fixing the six photosynthesis parameters to arbitrary values in their acceptable ranges. We run three Markov chains of 10 000 iterations with a five iterations thinning (i.e. we only consider one random sample out of five) to calibrate the parameters. During the second step, we fix the twelve carbon allocation parameters at the values obtained from the first step. We calibrate the six photosynthesis parameters by also running three Markov chains of 10 000 iterations with a five iterations thinning.




Finally, during the third step, the six photosynthesis parameters are fixed at the values obtained from the second step and the
twelve carbon allocation parameters are calibrated, by running three Markov chains of 30 000 iterations, with a five iterations
thinning as well. Each of those nine chains starts from random initial values of the parameters in their acceptable ranges. At
the end of each calibration step, we select the set of parameters with the highest posterior (Maximum A Posteriori value or
MAP, Hartig et al., 2019) from all iterations considering a burn-in period (i.e. the number of initial iterations of a chain that
are not considered in the calibration) of 1000 iterations (first and second steps) and 3000 iterations (third step). At the end of
the calibration process, we thus have six calibrated parameters from the second calibration step and twelve carbon allocation
parameters from the third one. The calibration method has been tested for convergence of Markov chains with Gelman-Rubin
statistical indicators (Hartig et al., 2019).

The MAIDEN model was calibrated at the twenty-one Eastern Canadian taiga sites and at the five aggregated sites over the
1950-2000 time period using the high- (NRCAN), mid- (GMF) and low-resolution (20CRv2c) datasets as inputs, as well as the
bias-corrected low-resolution dataset (20CRv2c corr.), and over the 1900-2000 time period using the 20CRv2c and 20CRv2c
corr. datasets as climatic inputs. MAIDEN was also calibrated at the three European sites using GHCN station data over
1950-2000 (FINL045; EALP; SWIT179), 1909-1944 (FINL045) and 1910-1949 (EALP and SWIT179). Pearson correlation
coefficients between observed TRW and simulated Dstem were computed, as well as the corresponding confidence level. To
compare observed and simulated tree-ring growth data after the optimization of the model parameters, both observed tree-ring
width series and simulated time series have been normalized to unitless indexes.

### 2.3.2    The VS-Lite model

The VS-Lite parameters are calibrated at each location following a bayesian approach described in Tolwinski-Ward et al.
(2013). The method is based on a standard Markov chain Monte Carlo approach, a Metropolis-Hastings algorithm embedded
within a Gibbs sampler. The VS-Lite model was calibrated at the same sites and over the same time periods as MAIDEN, using
the same climatic data (Sect. 2.3.1). Pearson correlation coefficients between TRW observations and simulated tree-growth
indexes were also computed. Observed time series have been normalized to unitless indexes as well.

### 2.4    Validation

Split-sample validation are performed by dividing the available data into two subperiods, one for calibration and one for
validation, and vice-versa. In order to test the influence of time series length, we validate the model for both short (1950-
1974 and 1975-2000) and long (1909-1944 and 1950-2000 or 1910-1949 and 1950-2000) time periods. For each validation
experiment, pearson correlation coefficients between observed TRW and simulated Dstem were computed, as well as the
corresponding confidence level.

Split-sample validation was preferred over other validation methods such as h-block Jack-knife which are computationally
intensive. Additionally, removing years may be inappropriate for the validation because of the autocorrelation charaterizing
yearly TRW observations. Similar problems arise from a bootstrap technique (Gea-Izquierdo et al., 2017).



**Table 3.** Description of each experiment performed in our study: experiment name; sites and climate dataset used for the experiment; time period of the experiment; short description of the experiment. Information on climate datasets can be found in Table 2. Individual and aggregated Eastern Canadian taiga sites refer to Fig. 1 and European sites refer to Fig. 2.

| Experiment name | Sites | Climate dataset | Time period | Description |
|---|---|---|---|---|
| **Calibration strategies for MAIDEN** | | | | |
| Application of prior MAIDEN parameters to all Canadian sites (Sect. 3.1) | Individual and aggregated Eastern Canadian taiga sites | NRCAN | 1950-2000 | We apply *QC_taiga* parameters as calibrated by Gennaretti et al. (2017) to all Eastern Canadian taiga sites |
| Site-specific calibration of the MAIDEN parameters and sensitivity to the quality of climatic inputs (Sect. 3.2) | Individual and aggregated Eastern Canadian taiga sites | NRCAN, GMF, 20CRv2c, 20CRv2c corr. | 1950-2000 ;1900-2000 (20CRv2c and 20CRv2c corr. only) | We calibrate each Eastern Canadian taiga sites with a bayesian procedure and evaluate the sensitivity of the calibration to the climate inputs quality |
| Regional calibration of MAIDEN (Sect. 3.3) | Individual and aggregated Eastern Canadian taiga sites | NRCAN | 1950-2000 | We evaluate the performance of MAIDEN at the Eastern Canadian taiga sites using a regional calibration |
| **Validation of MAIDEN** | | | | |
| Split-sample validation of MAIDEN calibration (Sect. 3.4) | Aggregated Eastern Canadian taiga sites (AC) and European sites (E) | NRCAN (AC); GHCN (E) | 1950-1974/1975-2000 (AC, E); 1909-1944 or 1910-1949/1950-2000 (E) | We validate our calibration procedure for MAIDEN using a split-sample method |
| **Comparison between models** | | | | |
| Comparison between VS-Lite and MAIDEN (Sect. 3.5) | Individual Eastern Canadian taiga sites (IC) and European sites (E) | NRCAN (IC); GHCN (E) | 1950-1974/1975-2000 (E); 1909-1944 or 1910-1949/1950-2000 (E); 1950-2000 (IC) | We compare VS-Lite and MAIDEN calibration and validation statistics |

## 3 Results and Discussion

Our results and discussion are structured into five sections that allow together to fulfil our objective of testing the applicability of MAIDEN over the twentieth century (Table 3). At first, we want to determine the best set of parameters for MAIDEN at our study sites and test the sensitivity of calibration to the quality of climatic inputs (Sect. 3.1, 3.2 and 3.3). In a context of





paleoclimate model-data comparison where MAIDEN will be driven by climate models outputs at low resolution, this is a crucial point of our analysis. For example, bias-correction and downscaling techniques could be good options to improve the robustness of the model calibration if the model is sensitive to the quality of climatic inputs.

We first test the possibility of using calibrated parameters from a well-documented site at other similar sites in terms of
environment (here, the Eastern Canadian taiga) and tree species (here, *Picea mariana*), in Sect. 3.1. Another option is to calibrate each site individually, as in Sect. 3.2 following the calibration protocol detailed in Sect. 2.3.1. We thirdly test in Sect. 3.3 an alternative calibration method which consists in calibrating the MAIDEN model over the mean of a tree-ring width observations network with similar environmental conditions and then applying the resulting calibrated parameters to the individual sites. From another perspective, this experiment could also be seen as an alternative method for the validation of the
MAIDEN model when the climate and/or tree-ring width observations time-series are too short for a split-sample validation. In this case, the individual sites are considered as nearly independent validation data. To test the sensitivity of the model to the quality of climatic inputs, we have selected four climatic datasets at different spatial resolution (Sect. 2.2.2, Table 2) that will be used in Sect. 3.2 to drive MAIDEN at the Eastern Canadian taiga sites. As a second sensitivity experiment, we have applied the parameters calibrated with MAIDEN using the high-resolution climatic data (NRCAN) to the Eastern Canadian taiga sites
driven by the low-resolution data without or with bias-correction (20CRv2c and 20CRv2c corr.).

The validation of MAIDEN in Sect. 3.4 is essential to evaluate the robustness of the calibration. The last section of our study consists in comparing the performance of the complex model MAIDEN with the performance of the simple model VS-Lite so as to assess the benefits of using a complex tree-growth model as MAIDEN for past climate reconstruction compared to a simple one (Sect. 3.5).

## 235  3.1  Application of prior MAIDEN parameters to all Canadian sites

At first, the *QC_taiga* parameters as calibrated by Gennaretti et al. (2017) (twelve carbon allocation and six photosynthesis parameters) were applied to the other twenty Eastern Canadian sites and five aggregated sites from Nicault et al. (2014) and Boucher et al. (2017) using the NRCAN (5') climate data (Table 2) over the 1950-2000 time period. Correlations between observations and simulations with MAIDEN using *QC_taiga* calibrated parameters (Fig. 3) are low and non-significant at
most sites. Several reasons can explain the low skill of MAIDEN using those parameters. These results could be linked to the lower replication level at the sites from Nicault et al. (2014) and Boucher et al. (2017) - even when aggregated - compared to the site from Gennaretti et al. (2017), that weakens the climatic signal in the series. This may also be due to a high sensitivity of parameters calibration to an unstable climate-species relationship among sites that are different from each other in many aspects (such as soil type, vegetation, nutrient availibility, for example). Additionally, the long-term trends of forest growth
in the Eastern Canadian taiga mostly depend on the past fire history (e.g. Payette et al., 2008; Gennaretti et al., 2014a; Erni et al., 2017). This represents the main natural disturbance factor that has shaped the North American boreal ecosystem by determining forest structure and composition as well as carbon stocks, and interacting with climate on a long time-scale. Yet, MAIDEN does not account for disturbances. To evaluate the effect of those disturbances on our experiment, the long-term decadal trends have been removed in both observations and simulations following Gennaretti et al. (2017) (Fig. S1). With




only the high frequency signal, the agreement between TRW observations and simulations with MAIDEN using *QC_taiga*
calibrated parameters is far better for most individual and aggregated sites.

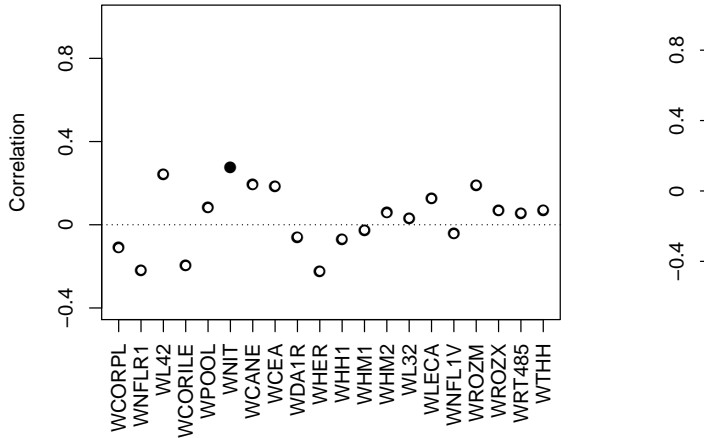
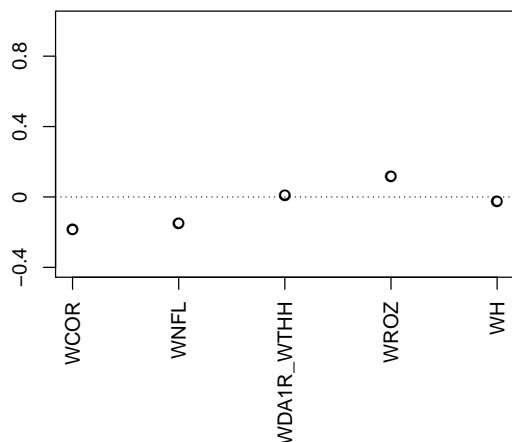

**Figure 3.** Pearson correlation coefficients between tree growth observations and simulations at the Eastern Canadian taiga sites (Fig. 1) with
MAIDEN using NRCAN (5') as climatic inputs (Table 2) for the 1950-2000 period with *QC_taiga* calibrated parameters from Gennaretti
et al. (2017). Individual (left) and aggregated sites (right). White inner circles stand for non-significant correlations (p-value > 0.05). Plain
circles stand for significant correlations (p-value < 0.05).

### 3.2 Site-specific calibration of the MAIDEN parameters and sensitivity to the quality of climatic inputs

A second option is to calibrate each of the twenty-one Eastern Canadian taiga sites as well as the five aggregated Eastern Cana-
dian taiga sites (Fig. 1) using the calibration procedure detailed in Sect. 2.3.1. Correlations between tree growth observations
and simulations with MAIDEN for the 1950-2000 calibration period at the Eastern Canadian taiga sites are good and significant
for all the climatic datasets (Fig. 4a). Correlations are in general slightly higher for the higher resolution datasets (NRCAN (5')
and GMF (1°) datasets, with an average correlation of 0.62 and 0.65 respectively compared with 0.57 for 20CRv2c (2°) and
0.61 for 20CRv2c corr. (2°)). At the aggregated sites (Fig. 5a), correlations for each dataset increase a little bit compared to
the average of individual correlations but the general picture is the same. The bias-correction (20CRv2c corr. (2°)) can slightly
improve correlations for the 20CRv2c (2°) climatic dataset in some cases (e.g. WL42 and WROZM). Consequently, those
results do not indicate that using higher resolution datasets increase effectively correlations. This is likely due to the calibra-
tion procedure that might be able to compensate for specific biases in each climatic dataset. This implies large variations of
calibrated parameters between experiments (Fig. S2 and S3), questionning the robustness of the selected values. The calibra-
tion method can also compensate potential biases of tree-ring observations and of sampling procedures which have important
impacts on long-term decadal trends (e.g. biases due to disturbance origin and tree selection criteria) (Johnson and Abrams,
2009; Gennaretti et al., 2014a; Duchesne et al., 2019).





Many potential biases of tree-ring observations due to the specific physiology of selected trees – that may not be representative of forest processes – and the chronology building process exist that may dampen the comparison with what MAIDEN simulates, i.e forest carbon accumulation and not forest demographic processes (Johnson and Abrams, 2009; Duchesne et al.,

2019). Ideally, considering those biases, we should find a better way to transform tree-ring data in time series with meaningful units to improve model-data comparisons. For example, Gennaretti et al. (2018) compute a wood biomass production index, which is closer to what MAIDEN simulates. This implies to have access to both tree-ring width and density measurements. However, given our global scale goals, this approach may be difficult to consider due to the lower availability of tree-ring density data (e.g. PAGES 2k Consortium, 2017).

Pearson correlations coefficients between TRW observations and tree-growth index simulations by MAIDEN for the 1900-2000 calibration period (Fig. 4b) are in most cases lower than those of the 1950-2000 calibration period. The bias-correction can slightly improve correlations in some cases but the latter stay smaller. At the aggregated sites (Fig. 5b), correlations for each dataset decrease slightly compared to the mean of individual correlations. The low correlation for the whole twentieth century can be explained by the large uncertainty of the 20CRv2c (2°) climatic dataset before 1950 there, as measured by the

large spread of the 20CRv2c ensemble spread at that time (Fig. S4).

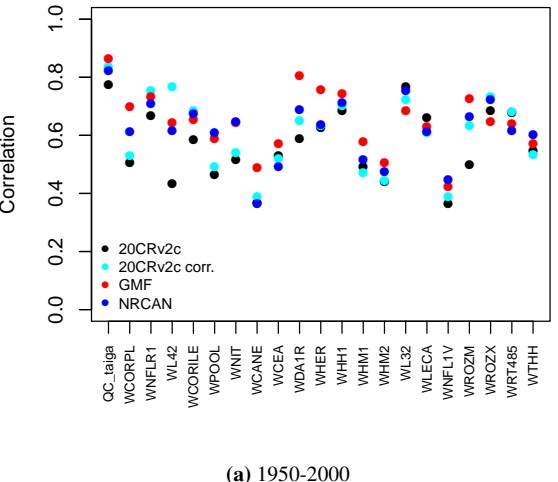

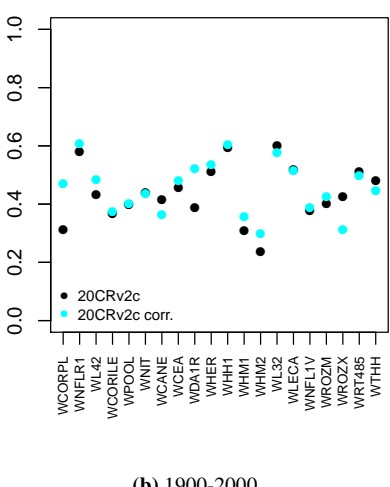

(a) 1950-2000                            (b) 1900-2000

**Figure 4.** Pearson correlation coefficients between tree growth observations and simulations at the Eastern Canadian taiga sites (Fig. 1a) with MAIDEN using the different climatic datasets described in Table 2 as inputs for the 1950-2000 (a) and 1900-2000 (b) calibration periods. White inner circles stand for non-significant correlations (p-value > 0.05). Here, all circles are plain because correlations are all significant.

When applying the parameters calibrated using the highest resolution dataset NRCAN (5') as climatic inputs to the Eastern Canadian taiga sites driven by 20CRv2c (2°) dataset (Fig. 6, right, in red), correlations are in average much lower. Mean correlation is 0.17 in that case compared to 0.57 when the parameters are calibrated using 20CRv2c (2°) as climatic inputs. With the 20CRv2c corr. (2°) dataset as climatic inputs – i.e. the low-resolution dataset corrected for bias in the mean seasonal

cycle – (Fig. 6, left, in red) we see that the performance of the MAIDEN model when applying NRCAN (5') parameters





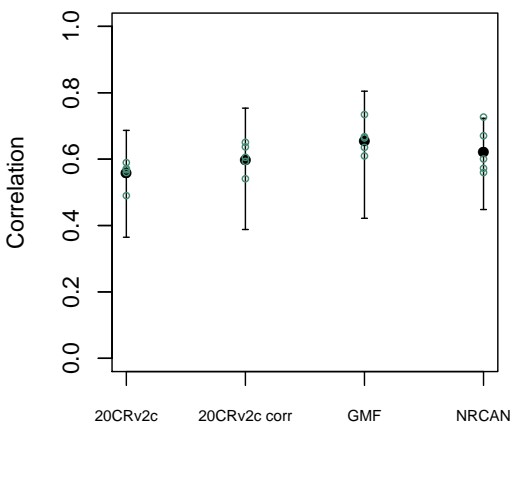

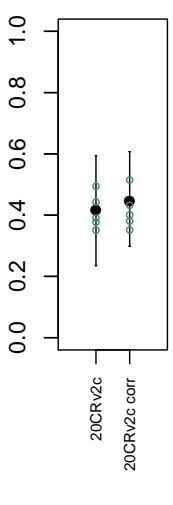

(a) 1950-2000    (b) 1900-2000

**Figure 5.** Pearson correlation coefficients (aggregated Eastern Canadian taiga sites (Fig. 1b), green circles), and mean and range of correlations (individual Eastern Canadian taiga sites used in aggregation (Fig. 1a and b), in black) between tree growth observations and simulations with MAIDEN using the different climatic datasets described in Table 2 as inputs for the 1950-2000 (a) and 1900-2000 (b) calibration periods.

is less good compared to the case when the parameters are calibrated using 20CRv2c corr. (2°) as climatic inputs (in black). Nevertheless, correlations are far better than with 20CRv2c (2°) (Fig. 6, right, in red). Indeed, the mean correlation is 0.36 when applying NRCAN (5') parameters and 0.61 when applying 20CRv2c corr. (2°) parameters. Consequently, the bias-correction of the 20CRv2c (2°) increases the robustness of the calibration of the MAIDEN parameters. Additionally, this shows that the

MAIDEN model parameters calibration is highly sensitive to the quality of the climatic dataset used as inputs.

At the aggregated sites (Fig. 7), the general picture is the same but with far lower correlations. The mean correlations are 0.07 when applying the parameters calibrated using NRCAN (5') to the aggregated sites driven by 20CRv2c (2°) dataset and 0.56 when the parameters are calibrated using 20CRv2c (2°). With the 20CRv2c corr. (2°) dataset as climatic inputs, mean correlations are respectively 0.18 and 0.61 with NRCAN (5') and 20CRv2c corr. (2°) parameters. Those results would require

a case-by-case analysis as it seems that higher replication does not provide better performance in this specific experiment.

### 3.3 Regional calibration of MAIDEN

At last, we apply the parameters calibrated against the mean of TRW observations from the twenty Eastern Canadian taiga sites (Fig. 8) to the five aggregated sites (Fig. 8, right) and to the individual sites used in the aggregation procedure (Fig. 8, left). For this experiment, we use the NRCAN (5') climate data (Sect. 2.2.2, Table 2) averaged over individual sites for each

aggregated site (Table 1). The main parameters linked to site conditions and control parameters (Table S1) are fixed to their mode (soil parameters), mean (site latitude, elevation and isohyet, station elevation and isohyet) or common value (*exp_site*,





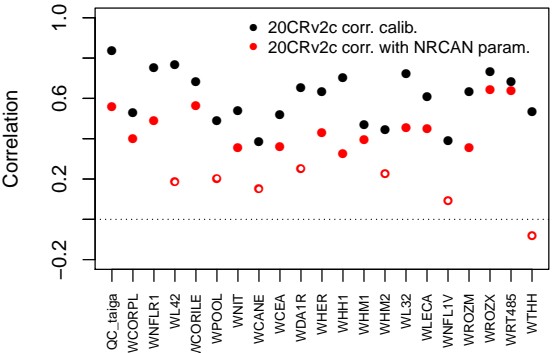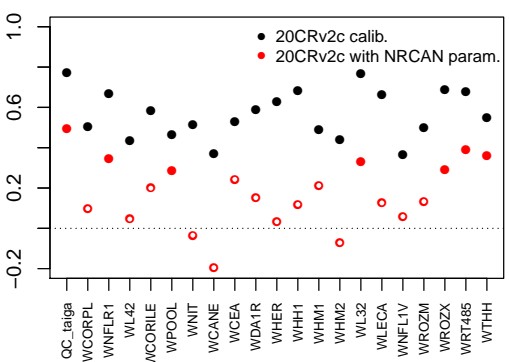

**Figure 6.** Pearson correlation coefficients between tree growth observations and simulations at the Eastern Canadian taiga sites (Fig. 1a) with MAIDEN using the 20CRv2c corr. (2°) (left) or 20CRv2c (2°) (right) climatic dataset for the 1950-2000 period with parameters calibrated using NRCAN (5') (with NRCAN param.) climatic inputs and with parameters calibrated using 20CRv2c corr. (2°) (left) or 20CRv2c (2°) (right) (calib.) climatic inputs (Table 2). White inner circles stand for non-significant correlations (p-value > 0.05).

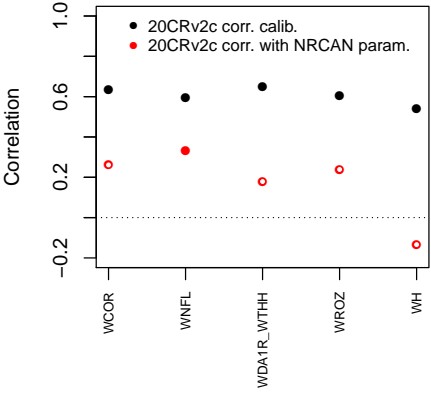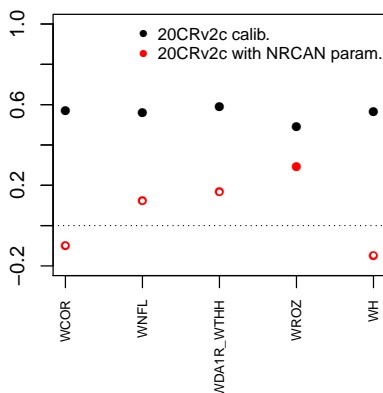

**Figure 7.** Pearson correlation coefficients between tree growth observations and simulations at the aggregated Eastern Canadian taiga sites (Fig. 1b) with MAIDEN using the 20CRv2c corr. (2°) (left) or 20CRv2c (2°) (right) climatic dataset for the 1950-2000 period with parameters calibrated using NRCAN (5') (with NRCAN param.) climatic inputs and with parameters calibrated using 20CRv2c corr. (2°) (left) or 20CRv2c (2°) (right) (calib.) climatic inputs (Table 2). White inner circles stand for non-significant correlations (p-value > 0.05).

slope and aspect parameters). Overall, correlations between TRW observations and simulations by MAIDEN with parameters calibrated based on the mean of the observed TRW time series are low and non-significant for the individual sites (Fig. 8, left). At the more replicated aggregated sites (Fig. 8, right), correlations between TRW observations and simulations get better with
three significant correlations out of five sites. However, this result should be viewed in parallel with the individual correlations (Fig. 8, left) and sites implied in the aggregation (Table 1). Indeed, aggregated sites with higher correlations are made up of individual sites with higher correlations as well. It means that probably not only higher replication is at the origin of higher





correlations for most aggregated sites but also the specific conditions at each individual site as well as site ecological history, as previously mentioned (Sect. 3.1).

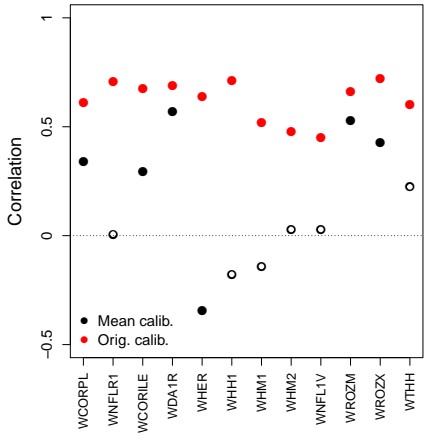 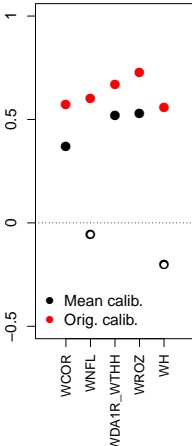

**Figure 8.** Pearson correlation coefficients between tree growth observations and simulations at the individual (left) and aggregated Eastern Canadian taiga sites (right) (Fig. 1a and b) with MAIDEN using the NRCAN (5') climatic dataset (Table 2) with site-specific calibration of the parameters (Orig. calib., in red) and with parameters calibrated based on the mean of the observed TRW time series (Mean calib.) for the 1950-2000 period. White inner circles stand for non-significant correlations (p-value > 0.05).

### 3.4 Split-sample validation of MAIDEN calibration

Depending on the available years, we have selected different time periods at the European sites (Table 4) and at the aggregated Eastern Canadian taiga sites (Table 5), using each period once for the calibration and once for the validation. At the European sites, twenty-five years is clearly a too short period of time to get robust results while the validation is generally successful for the longer period as indicated by the significant correlations – except in one case – (Table 4). Similarly, at the aggregated Eastern Canadian sites – where we only have fifty years of reliable climate data (see Sect. 3.2) – , a twenty-five years subperiod is not enough for a robust calibration and validation (Table 5). However, even on the long time period (Table 4), we can see a clue of some overfitting, especially at the SWIT179 site, where the correlation for the validation period is far lower compared to the correlation for the calibration period. Those results show that because of the large number of parameters, the validation of MAIDEN is difficult. It requires long observation series but the skill of the model still decreases significantly for the validation period.

### 3.5 Comparison with VS-Lite

In average, over the 1950-2000 calibration period at the individual Eastern Canadian taiga sites, VS-Lite has lower correlations for the highest resolution dataset (NRCAN) compared with MAIDEN, i.e. 0.106 and 0.62 mean correlations for VS-Lite and MAIDEN respectively (Fig. 9). Results for the other climatic datasets over the 1950-2000 (GMF (1°), 20CRv2c (2°) and



**Table 4.** Pearson correlation coefficients between tree growth observations and simulations at the European sites (Fig. 2) with MAIDEN and VS-Lite using GHCN as climatic inputs (Table 2) for the 1950-1974 and 1975-2000 and for the 1910-1949 (EALP, SWIT179) or 1909-1944 (FINL045) and 1950-2000 calibration and validation periods and vice-versa. Asterisks stand for significant correlations (p-value < 0.05).

| European sites | Model | 1950-1974 | | 1975-2000 | |
|---|---|---|---|---|---|
| | | Calibration | Validation | Calibration | Validation |
| EALP | MAIDEN | 0.831* | 0.443* | 0.886* | 0.546* |
| | VS-Lite | 0.629* | 0.618* | 0.603* | 0.599* |
| SWIT179 | MAIDEN | 0.744* | 0.284 | 0.783* | 0.325 |
| | VS-Lite | 0.260 | 0.181 | 0.435* | 0.396* |
| FINL045 | MAIDEN | 0.827* | 0.0358 | 0.610* | 0.135 |
| | VS-Lite | 0.415* | 0.209 | 0.271 | 0.143 |

| | | 1910-1949 or 1909-1944 | | 1950-2000 | |
|---|---|---|---|---|---|
| | | Calibration | Validation | Calibration | Validation |
| EALP | MAIDEN | 0.880* | 0.626* | 0.856* | 0.569* |
| | VS-Lite | 0.491* | 0.487* | 0.656* | 0.656* |
| SWIT179 | MAIDEN | 0.721* | 0.163 | 0.659* | 0.306* |
| | VS-Lite | 0.490* | 0.489* | 0.350* | 0.353* |
| FINL045 | MAIDEN | 0.751* | 0.428* | 0.670* | 0.394* |
| | VS-Lite | 0.320 | 0.304 | 0.315* | 0.263 |

20CRv2c corr. (2°)) and over the 1900-2000 calibration periods (20CRv2c (2°) and 20CRv2c corr. (2°) climatic datasets) also show lower correlations compared to MAIDEN (Fig. S5). As for split-sample validation over the long time period, the performance of VS-Lite is more stable (less fall of validation than calibration correlation) compared with MAIDEN (Table 4) even if correlations are, except for SWIT179, lower than MAIDEN. Similarly, over the short time period, the performance of VS-Lite is less good than over the long time period but still more stable than MAIDEN (Table 4). Compared to VS-Lite,

MAIDEN has shown lower skill over short time period validation that indicates that we should only use MAIDEN when a long enough period is available for validation. As for long validation period, MAIDEN has shown a stronger decrease in correlations compared to VS-Lite but still with higher correlations than VS-lite on average. This would indicate that MAIDEN calibration is not always prone to overfitting.




**Table 5.** Pearson correlation coefficients between tree growth observations and simulations at the aggregated Eastern Canadian sites (Fig. 1b) with MAIDEN using NRCAN (5') as climatic inputs (Table 2) for the 1950-1974 and 1975-2000 calibration and validation periods and vice-versa. Asterisks stand for significant correlations (p-value < 0.05).

| Canadian sites | 1950-1974 | | 1975-2000 | |
|---|---|---|---|---|
| | Calibration | Validation | Calibration | Validation |
| WCOR | 0.693* | 0.146 | 0.783* | 0.589* |
| WNFL | 0.619* | 0.103 | 0.804* | 0.429* |
| WDA1R_WTHH | 0.480* | 0.737* | 0.610* | 0.332 |
| WROZ | 0.674* | 0.577* | 0.841* | 0.270 |
| WH | 0.549* | 0.008 | 0.718* | -0.011 |

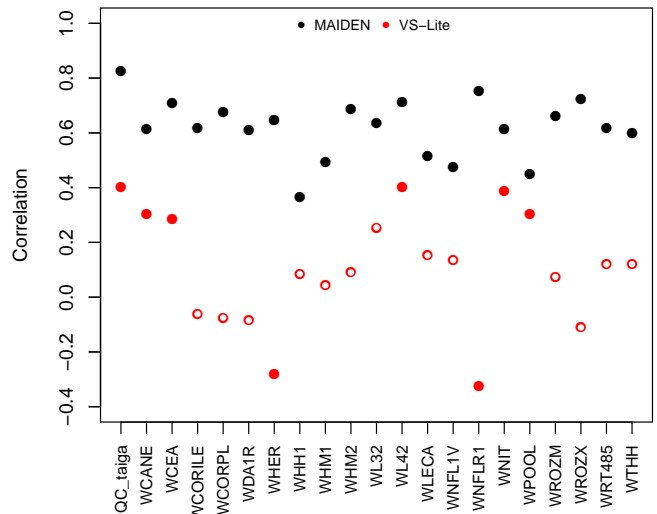

**Figure 9.** Pearson correlation coefficients between tree growth observations and simulations at the Eastern Canadian taiga sites (Fig. 1a) with VS-Lite (in red) and MAIDEN (in black) using NRCAN (5') as climatic inputs (Table 2) for the 1950-2000 calibration period. White inner circles stand for non-significant correlations (p-value > 0.05).

## 4 Conclusions

In this paper we have tested the applicability of the ecophysiological tree-growth model MAIDEN for potential dendroclimatological applications during the twentieth century at twenty-one Eastern Canadian taiga sites and three European sites using tree-ring width observations. Our results provide a protocol for the application of MAIDEN to potentially any site with tree-ring width data in the extratropical region, from climatic data selection to validation step, through automatised bayesian calibration





of the most sensitive parameters. As the ultimate goal is to use MAIDEN in a context of paleoclimatic reconstruction, forced

by low-resolution climate models outputs, we also analysed the sensitivity of the model to parameters calibration and to the quality of climatic inputs. The performance of MAIDEN was compared to the one of a simple tree-growth model, VS-Lite, to evaluate the advantages of using a complex tree-growth model for past climate reconstruction.

Different strategies have been tested to select the value for the most sensitive parameters of the MAIDEN model. When applying calibrated parameters from a well-documented site at other sites with same species and similar environmental condi-

tions, very low correlations between tree-ring width observations and simulations by the MAIDEN model are found. However, when removing the long-term trend to account for the past disturbance-history of these sites that is not represented in MAIDEN, correlations get higher. In the future, this strategy can be used by selecting sites carefully to avoid disturbances. At our study sites, the bayesian calibration of the most sensitive parameters of the model can provide good and significant correlations between tree-growth observations and simulations.

Secondly, sensitivity of the MAIDEN model parameters calibration to the quality of the climatic data used as inputs has been highlighted. In a context of paleoclimatic applications, where MAIDEN will be used driven by climate models outputs at low resolution, bias-correction and downscaling techniques could be good options to improve climate inputs and calibration quality, leading thereby to reasonable correlations with observed tree-ring width.

Our split-sample validation experiments are encouraging. However, when a calibration interval of only a few decades is

available, the calibration display large overfitting for individual sites as indicated by the very low correlation with observations over the validation period. Similar split-sample experiments on longer series show much better results, with potentially some overfitting but still with relatively high and generally significant correlations over the validation period. When working with a network of similar sites, the alternative validation technique, i.e. applying calibrated parameters from the mean of a network of tree-ring width observations series with same species and environmental conditions to the individual sites, should be preferred

if not enough data (climate and TRW observations) are available for split-sample validation.

Lastly, at our study sites, MAIDEN has shown higher calibration and validation correlations in most cases compared to VS-Lite. VS-Lite correlations over the calibration period are especially far lower for sites with low replication (i.e. the Eastern Canadian taiga sites from Nicault et al. (2014) and Boucher et al. (2017)). However, VS-Lite stays more stable over both calibration and validation periods. Consequently, VS-Lite has a lower ability to reproduce tree growth at our sites but is prone

to a lower risk of overfitting than MAIDEN. Most importantly, we have shown that, to limit overfitting, MAIDEN should not be used with short and low-replicated tree-ring width observations time series. VS-Lite is less risky to use in such situations as there is potentially less overfitting in the calibration and probably easier to apply over a large network of tree-ring width time series. However, VS-Lite does not include neither $CO_2$ nor biological processes and may thus not be able to take into account changes in conditions between the recent calibration period and the more distant past.

In the future, MAIDEN will be applied at a larger spatial scale in a systematic way using the protocol that has been developed here, by selecting hundreds of sites from the commonly used databases in paleoclimate reconstruction based on tree-ring proxies, covering a wide range of environmental conditions and tree species, such as PAGES2k (PAGES 2k Consortium, 2017) and NTREND (Anchukaitis et al., 2017; Wilson et al., 2016). This broader analysis will allow us to refine the protocol



developed here in order to identify the sites where MAIDEN can be successfully applied and estimate the uncertainty associated
with the use of MAIDEN for many more different sites.

Although some limitations could remain in our calibration protocol, we have shown the ability of MAIDEN to simulate tree-growth index time series that can fit robustly tree-ring width observations under certain conditions (well-replicated tree-ring width observations time series, high-resolution or downscaled climate data, long time period), as well as its potential to be used as a complex mechanistic proxy system model in paleoclimatic applications and more specifically in data assimilation.

*Competing interests.* The authors declare that there is no conflict of interest.

*Acknowledgements.* JR is F.R.S-FNRS Research Fellow, Belgium; HG is Research Director at F.R.S.-FNRS, Belgium; JG is Research Director at CNRS, France. This publication has received partial funding from Laboratory of Excellence OT-Med (project ANR-11- LABEX-0061, A\*MIDEX project 11-IDEX-0001-02) as well as from F.R.S-FNRS and Fonds de recherche Société et culture Québec through the ClimHuNor project.





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
