# Peer review of "Application and evaluation of the dendroclimatic process-based model MAIDEN during the last century in Canada and Europe"

_Climate of the Past, 2019_

## Referee Comment (RC1) · Vladimir Shishov (Referee) · 31 Dec 2019

General comment

The paper "Application and evaluation of the dendroclimatic process-based model MAIDEN during the last century in Canada and Europe" by Rezsöhazy et al. is a good example to explain specifics of MAIDEN model application taking into account a complexity of such multidimensional tool to simulate tree growth under climatic influence in different environments.

The overall impression of the paper is very good. The logical structure of the

manuscript, a detailed description of the parameterization procedure of the model itself and skills comparison of two models: VS-Lite and MAIDEN are noteworthy. I want to underline that the parametrization of such models, their calibration and verification is a key point to apply correctly a tree-growth simulation in different habitats.

The authors mentioned that their "results provide a protocol for the application of MAIDEN to potentially any site with tree-ring width data in the extratropical region". I am wondering did the authors make the MAIDEN code available in some open-access depository to use it for wider group of researchers. I am sure the tables of optimal parameter values for some sites as well as corresponded climate data and tree-ring chronologies putting on-line will allow to make the model itself more applicable in the research community.

I suggest that the paper can be published after minor revision.

Specific comments

Section 100 "…the ongoing phenological phase (five phases per year: winter 1, winter 2, budburst, summer and fall)" Could the authors explain what is the difference between winter1 and winter 2 phenological phases?

Section 125 "Those chronologies have been standardized using the Age-Band Regional Curve Standardization (or RCS) method". Did the authors use pith estimations for individual tree-ring series? Did the authors split fast and slow growing trees to avoid end-effect bias?

Section 135 "…we get five aggregated sites (Table 1)" What are intersite correlations (Rbar) between tree-ring chronologies at the same one-degree grid? Could the authors clarity this point in the paper?

Section 135 "This observational network represents an archetypal example of a singular species that covers an important hydroclimatic gradient" Why is the gradient important? Could the authors explain it?

Section 140 "...standardized with a cubic-smoothing spline with a 50% frequency cut-off at 35 years;" and "... standardized using a spline with a 50% frequency cut-off response at 32 years". The authors mentioned that the European sites has a good replication of wooden samples which is a necessary condition to apply the same standardization strategy as for Canada. What was a reason to use another standardization technique for Europe which could be a reason of the end-effect bias (Melvin, 2004)?

Section 170 "The comparison relies on the computation of the model likelihood defined as the sum of the logarithms of the normal probability densities of the residuals between the model simulation and the observations". Why the authors use the logarithms of the normal probability densities of the residuals? Are the residuals non-normal distributed? It seems to me by such transformation the authors tried to adopt the Markov chains procedure to their parametrization taking into account strong requirement of data normality in Markov processes.

Section 190 "Pearson correlation coefficients between observed TRW and simulated Dstem were computed, as well as the corresponding confidence level" Pearson correlation is not enough to guarantee a convergence of simulated curve with initial chronology. Why did not the authors use an additional criterion such as RMSE minimising or others?

Section 200 "The VS-Lite parameters are calibrated at each location..." How many parameters were optimized keeping in mind that overall 11 of them were used in the VS-lite? Could the authors describe them more precisely in the ms.

Supplementary materials. Could the authors include the table with the optimal MAIDEN and VS-lite parameter values for all sites in Canada and Europe?

Supplementary materials. Among with Fig. S2, S3 could the authors include the obtained distributions of the MAIDEN parameters?

Supplementary materials. Could the authors include the obtained distribution of the

VS-lite parameters?

Please also note the supplement to this comment:
https://www.clim-past-discuss.net/cp-2019-140/cp-2019-140-RC1-supplement.pdf

—————————————————————

---

## Referee Comment (RC2) · Anonymous Referee #2 · 30 Mar 2020

Review of CP-2019-140

This manuscript presents a useful analysis of the use of the model MAIDEN as a PSM for potential paleoclimatic reconstructions. I have some minor comments, corrections, and requests for clarification.

I think it would be important to state more prominently that the results here come with the caveat that they are done over a limited range of climate regimes. In my experience using VS-lite, I have found large differences for Eastern North America vs. Western North America, where Eastern North America (the primary region used here) did clearly worse than Western North America. It's therefore possible that MAIDEN will be

less clearly the winner in certain climate regimes.

All of the validations are done with only the correlation metric. Correlation will miss potentially important differences like a variance bias. Is this not a concern here because the time series being compared are all standardized to have no mean and unit variance?

I'm confused about the use of NRCAN data in the VS-lite model. If I've understood the manuscript correctly, the NRCAN data provides daily max-min temperature and precip data. But I believe that VS-lite is designed for monthly mean data. Is NRCAN (and daily max/min values) the right data to be using for VS-lite? I'm wondering if this might contribute to the reduced skill of VS-lite.

Can the authors comment on the computation cost of running MAIDEN vs VS-lite? This is particularly relevant for paleoclimate DA where an expensive PSM might be justification enough for not using it if something else is much faster.

p2.l51-53 This isn't actually true. Several reconstructions have assimilated proxy values directly using linear statistical "PSMs" (e.g., Hakim et al. 2016, Steiger et al. 2018, Tardif et al. 2018). While these are not physically-based, they still are a kind of PSM and the proxy values are not converted to temperature and then assimilated. Additionally there are reconstructions methods that have tested the direct assimilation of real isotope data using isotope GCMs (Steiger et al. 2017, Okazaki and Yoshimura 2019), and thus employed fully physically-based PSMs.

p3.l62-64 Is the inclusion of $CO_2$ influences needed for Common Era paleoclimate though? Over most of the Common Era $CO_2$ changes very little. Then when $CO_2$ does start to matter, we have plentiful observations? Maybe there's some other aspect of the MAIDEN model that would be more beneficial to highlight for paleoclimatic applications? It just seems like the use of MAIDEN might not be sufficiently motivated here.

---

## Referee Comment (RC3) · Vladimir Shishov (Referee) · 27 Apr 2020

The authors carefully revised the MS following all my comments and suggestions step-by-step. I find that a new version of the paper can be published as it is.

---

## Author Comment (AC3) · 27 Apr 2020

Dear Referee,

We would like to warmly thank you for your positive feedback.

Kind regards,

On behalf of all Co-Authors,

Rezsöhazy Jeanne

---

## Author Response (AR2)

Dear Editor,

We would like to warmly thank you for editing our manuscript. We have updated it according to the comments of the reviewers as suggested in our responses. You can find here below the point-by-point response to reviewers comments, where the lines correspond to the revised manuscript and where slight changes in wording have been made, as well as the final version of our manuscript with track changes.

On behalf of all Co-Authors,

Jeanne Rezsöhazy

Dear editor and reviewers, we would like first to thank you for your useful feedbacks and comments on our manuscript. You can find here below the Referee's comments in *italics* and our answer in blue. In **bold**, you can find the modifications that will be made to the manuscript.

**Referee#1 Vladimir Shishov**

*The paper "Application and evaluation of the dendroclimatic process-based model MAIDEN during the last century in Canada and Europe" by Rezsöhazy et al. is a good example to explain specifics of MAIDEN model application taking into account a complexity of such multidimensional tool to simulate tree growth under climatic influence in different environments. The overall impression of the paper is very good. The logical structure of the manuscript, a detailed description of the parametrization procedure of the model itself and skills comparison of two models: VS-Lite and MAIDEN are noteworthy. I want to underline that the parametrization of such models, their calibration and verification is a key point to apply correctly a tree-growth simulation in different habitats.*

We would like to warmly thank the Referee for this very positive general feedback, for the careful evaluation of our manuscript as well as for the useful comments that will be addressed in the revised version as specified here below.

*The authors mentioned that their "results provide a protocol for the application of MAIDEN to potentially any site with tree-ring width data in the extratropical region". I am wondering did the authors make the MAIDEN code available in some open-access depository to use it for wider group of researchers. I am sure the tables of optimal parameter values for some sites as well as corresponded climate data and tree-ring chronologies putting on-line will allow to make the model itself more applicable in the research community.*

*I suggest that the paper can be published after minor revision.*

We agree with the Referee that an open-access depository with results and data from the paper would be worthwhile. Currently, all climatic data are publicly available (except NRCAN that is available on request) and the links for downloading them will be added to the manuscript. The links to access the European tree-ring width data will also be added. For the Eastern Canadian taiga sites from Nicault et al. (2014) and Boucher et al. (2017) that has been used in the paper, an online reference will be provided in the paper, that links to a web site under development to share the tree-ring network of Québec-Labrador from which the Canadian data in the manuscript come: http://dendro-qc-lab.ca/trw.html. Finally, the parameters values will be added in the supplementary material, following to another comment from the Referee (see below).

The structure of the MAIDEN model is visible online (https://figshare.com/articles/MAIDEN_ecophysiological_forest_model/5446435/1) and its modules are available upon request.

***Specific comments***
*Section 100 "...the ongoing phenological phase (five phases per year: winter 1, winter 2, budburst, summer and fall)" Could the authors explain what is the difference between winter1 and winter 2 phenological phases?*

The explanation will be added to the text on lines 103-104 (p.4), as follows: "(five phases per year: **winter 1 with no accumulation of growing degree days (GDD), winter 2 with active GDD accumulation**, budburst, summer and fall)".

*Section 125 "Those chronologies have been standardized using the Age-Band Regional Curve Standardization (or RCS) method". Did the authors use pith estimations for individual tree-ring series? Did the authors split fast and slow growing trees to avoid end-effect bias?*

We would like to highlight that the tree-ring series were compiled before this article. All trees were dated and measured on cross-sections sampled at breast height (1.3m). The pith offset was done one for all trees. All samples were collected on dominant trees growing in homogeneous forests and it was not necessary to separate fast-growing trees from slow growing trees in such conditions.

Accordingly, we will add the following information to the manuscript, on lines 129-133 (p.5): "A network of tree-ring width chronologies of Picea mariana collected in similar conditions is available for the Eastern Canadian taiga (Nicault et al., 2014; Boucher et al., 2017, http://dendro-qc-lab.ca/trw.html). **We use here the tree-ring series directly derived from this compilation, without any modification.** The chronologies have been **previously** standardized using the Age-Band Regional Curve Standardization (or RCS) method **proposed by Briffa et al. (2001) and further applied to a similar boreal dataset by Nicault et al. (2014)."**

Similarly, the same information will be added on lines 160-161 (p.6) for the European sites: "**Similarly to the Eastern Canadian taiga chronologies, the tree-ring series were not modified here."**

*Section 135 "...we get five aggregated sites (Table 1)" What are intersite correlations (Rbar) between tree-ring chronologies at the same one-degree grid? Could the authors clarity this point in the paper?*

Proximity between sites was used as a criterion for building our aggregated chronologies because we assume that we can reduce the non-climatic noise in low-replicated chronologies by averaging close chronologies. A one degree grid appears to us as an objective way to merge sites together. The intersite correlations between tree-ring chronologies (chronologies inside the same one-degree grid have the same colour) is presented here below (all significant at a confidence level of 99%).

The average intersite correlations for all aggregated sites will be added to the manuscript on lines 143-144 (p.5), as follows: "**The aggregation allows us to get relatively good inter-sites correlations inside the same one-degree grid, ranging from 0.442 to 0.732 with an average of 0.558."**.

|         | WCORPL | WNFL1V | WNFLR1 | WDA1R | WTHH  | WROZM | WROZX | WHER  | WHH1  | WHM1  | WHM2  |
|---------|--------|--------|--------|-------|-------|-------|-------|-------|-------|-------|-------|
| WCORILE | 0.692  | 0.539  | 0.624  | 0.626 | 0.643 | 0.653 | 0.588 | 0.472 | 0.374 | 0.586 | 0.461 |
| WCORPL  |        | 0.492  | 0.577  | 0.537 | 0.329 | 0.497 | 0.731 | 0.504 | 0.587 | 0.581 | 0.570 |
| WNFL1V  |        |        | 0.509  | 0.235 | 0.239 | 0.466 | 0.400 | 0.241 | 0.135 | 0.280 | 0.456 |
| WNFLR1  |        |        |        | 0.541 | 0.177 | 0.541 | 0.662 | 0.389 | 0.313 | 0.429 | 0.333 |
| WDA1R   |        |        |        |       | 0.442 | 0.621 | 0.586 | 0.303 | 0.548 | 0.579 | 0.493 |
| WTHH    |        |        |        |       |       | 0.494 | 0.140 | 0.222 | 0.025 | 0.535 | 0.296 |
| WROZM   |        |        |        |       |       |       | 0.582 | 0.331 | 0.349 | 0.598 | 0.499 |
| WROZX   |        |        |        |       |       |       |       | 0.548 | 0.641 | 0.528 | 0.518 |
| WHER    |        |        |        |       |       |       |       |       | 0.485 | 0.501 | 0.454 |
| WHH1    |        |        |        |       |       |       |       |       |       | 0.589 | 0.593 |
| WHM1    |        |        |        |       |       |       |       |       |       |       | 0.732 |
| WHM2    |        |        |        |       |       |       |       |       |       |       |       |

*Section 135 "This observational network represents an archetypal example of a singular species that covers an important hydroclimatic gradient" Why is the gradient important? Could the authors explain it?*

Sites located along the western (near James Bay, WNFL1V) and eastern (near Labrador sea, WL32) margins of the study area present the warmest growing seasons in the network (864 growing degree-days >5°C for the 1976-2005 period, Hutchinson et al., 2009). Sites located in the center of the Quebec-Labrador peninsula (WHM2) present a much shorter growing season (692 growing degree-days >5°C) much like the sites located further north (WLECA, 573 growing degree-days >5°C). Annual precipitation increase from west to east, passing from 668 mm (WNFL1V) to 907 mm (WL32) but significantly decrease with latitude, reaching only 567 mm (WLECA) for the 1976-2005 period (Hutchinson et al, 2009).

The manuscript will be modified accordingly on lines 144-152 (p.5), as follows: "This observational network represents an archetypal example of a singular species that covers an important hydroclimatic gradient. **Sites located along the western (near James Bay, WNFLV1, Fig. 1a) and eastern (near Labrador sea, WL32, Fig. 1a) margins of the study area present the warmest growing seasons in the network (864 growing degree-days above 5° for the 1976-2005 period, Hutchinson et al., 2009). Sites located in the center of the Quebec-Labrador peninsula (WHM2, Fig. 1a) present a much shorter growing season (692 growing degree-days above 5°), much like the sites located further north (WLECA, Fig. 1a, 573 growing degree-days above 5°). Annual precipitation increases from west to east, passing from 668 mm (WNFLV1, Fig. 1a) to 907 mm (WL32, Fig. 1a), and significantly decreases with latitude, reaching only 567 mm at WLECA (Fig. 1a) for the 1976-2005 period (Hutchinson et al., 2009). This makes it a relevant candidate for our calibration and validation exercises.**"

*Section 170 "The comparison relies on the computation of the model likelihood defined as the sum of the logarithms of the normal probability densities of the residuals between the model simulation and the observations". Why the authors use the logarithms of the normal probability densities of the residuals? Are the residuals non-normal distributed? It seems to me by such transformation the authors tried to adopt the Markov chains procedure to their parametrization taking into account strong requirement of data normality in Markov processes.*

The logarithmic transformation appears to us as a common operation to maximise likelihood in Bayesian statistics for reasons of algebraic simplicity as well as numerical stability, as mentioned in Vrugt (2016, p.275, just before equation (8)). This paper also presents the DREAM software that we use for the Bayesian calibration of our selected parameters.

Vrugt, J.A.: Markov chain Monte Carlo simulation using the DREAM sofware package: Theory, concepts, and MATLAB implementation, Environmental Modelling & Software, 75, 273-316, 2016.

*Section 190 "Pearson correlation coefficients between observed TRW and simulated Dstem were computed, as well as the corresponding confidence level" Pearson correlation is not enough to guarantee a convergence of simulated curve with initial chronology. Why did not the authors use an additional criterion such as RMSE minimising or others?*

We agree with the Referee that other indicators could have been used for the analysis. We wanted to only use one indicator in order to simplify the message but in the future, other statistical measures could be considered for a more careful evaluation of our method. We also would like to highlight that because of the normalization of both observations and simulations (due to different units), some indicators like RMSE do not bring much new information compared to correlations.

*Section 200 "The VS-Lite parameters are calibrated at each location..." How many parameters were optimized keeping in mind that overall 11 of them were used in the VS-lite? Could the authors describe them more precisely in the ms.*

Four VS-Lite parameters, corresponding to the lower and upper temperature ($T_1$ and $T_2$ in Tolwinski-Ward et al., 2011) and soil moisture ($M_1$ and $M_2$ in Tolwinski-Ward et al., 2011) thresholds of the model, have been optimized using the Matlab code from Tolwinski-Ward et al. (2013). The other parameters have been kept to default values. This information will be added to the manuscript on lines 224-227 (p.10), as follows: "The VS-Lite parameters are calibrated at each location following a bayesian approach described in Tolwinski-Ward et al. (2013). **In this study, four VS-Lite parameters, corresponding to the lower and upper temperature (respectively $T_1$ and $T_2$ in Tolwinski-Ward et al., 2011) and soil moisture (respectively $M_1$ and $M_2$ in Tolwinski-Ward et al., 2011) thresholds of the model, have been optimized. The other parameters were fixed to default values**."

*Supplementary materials. Could the authors include the table with the optimal MAIDEN and VS-lite parameter values for all sites in Canada and Europe?*

We will add the tables in the supplementary materials for all 21 Canadian sites, 5 aggregated Canadian sites and three European sites (1950-2000).

*Supplementary materials. Among with Fig. S2, S3 could the authors include the obtained distributions of the MAIDEN parameters?*

We will add the figures in the supplementary materials for all 21 Canadian sites, 5 aggregated Canadian sites (NRCAN high-resolution dataset) and three European sites (GHCN high-resolution dataset). The high-resolution dataset is the most relevant considering our results and adding more distributions to the supplementary materials will result in a high number of pages.

*Supplementary materials. Could the authors include the obtained distribution of the VS-lite*

*parameters?*

For technical reasons, and as the paper focusses on MAIDEN, we are unfortunately not able to provide the distributions that would correspond to several additional figures in an already long supplement.

Dear editor and reviewers, we would like first to thank you for your useful feedbacks and comments on our manuscript. You can find here below the Referee's comments in *italics* and our answer in blue. In **bold**, you can find the modifications that will be made to the manuscript.

**Referee#2**

*This manuscript presents a useful analysis of the use of the model MAIDEN as a PSM for potential paleoclimatic reconstructions. I have some minor comments, corrections, and requests for clarification.*

We would like to deeply thank the Referee for the positive evaluation of our manuscript and the interesting comments. They will all be accounted for in the revised manuscript, as described here below.

*I think it would be important to state more prominently that the results here come with the caveat that they are done over a limited range of climate regimes. In my experience using VS-lite, I have found large differences for Eastern North America vs. Western North America, where Eastern North America (the primary region used here) did clearly worse than Western North America. It's therefore possible that MAIDEN will be less clearly the winner in certain climate regimes.*

We totally agree with the reviewer that the results come with the caveat that they are done over a limited range of climate regimes and that an analysis on a broader scale is needed to have a complete view on the performance of both models under various climate conditions. The objective here is clearly not to present an exhaustive evaluation of the two models or of our calibration method but to test our methodology on a few sets of tree-ring sites with different configurations (a network and few individual sites in Europe), so as to present our methodology. We are currently testing the methodology exemplified in the manuscript to a wider range of environmentally different sites to test the applicability of our calibration method for the MAIDEN model.

Therefore, we will state this again on lines 366-369 (p.18), as follows: "**As our objective is to provide a first test of our calibration methodology using only a few sets of tree-ring sites, the obtained results only give an incomplete view of the MAIDEN model performance and its comparison with VS-Lite, focussing over a limited range of climate regimes. More experiments in different conditions are required in the future to exhaustively evaluate and compare the performance of both models.**"

*All of the validations are done with only the correlation metric. Correlation will miss potentially important differences like a variance bias. Is this not a concern here because the time series being compared are all standardized to have no mean and unit variance?*

We agree with the reviewer that our analysis do not allow estimating the variance bias. Ideally, an exhaustive quantitative evaluation of MAIDEN would require a comparison of the variable simulated by MAIDEN to represent tree-growth (which is the annual quantity of carbon allocated to the stem in $gC.m^2$ of forest per year) directly with observations. In this case, all biases (including on the variance) can be estimated. Unfortunately, this would, for example, imply to have observations such as tree-ring density measurements, which are less widely distributed than tree-ring widths, and to account for biases in tree-ring observations due to the chronology building process. Those biases may indeed deteriorate the comparison with what MAIDEN simulates, i.e forest carbon accumulation and not tree-ring indexes. In specific cases, we are able to compare outputs variables from MAIDEN with observations, as it is the case for example for simulated gross primary

production with eddy covariance stations measurements of gross ecosystem production (Gennaretti et al., 2017) but this is not possible in most paleoclimate applications.

Consequently, such as VS-Lite which produces a unitless tree-growth index, we have to use a simple normalization procedure, assuming that annual quantity of carbon allocated to the stem is proportional to tree-ring width observations, as stated in our original manuscript on lines 106-108 (p.4). The disadvantage is that this normalization forbids us to assess error in the variance. This is why we only analyse the correlations for simplicity as using other metrics like the RMSE would not help us in this aspect. Similarly, studies on VS-Lite such as Breitenmoser et al. (2014) or Tolwinski-Ward et al. (2011) have used correlation as a unique statistical indicator.

This will be mentioned more explicitly in the revised version of the manuscript on lines 215-222 (p.9-10):"To compare observed and simulated tree-ring growth data after the optimization of the model parameters, both observed tree-ring width series and simulated time series have been normalized to unitless indexes. **Ideally, an exhaustive quantitative evaluation of MAIDEN would require a comparison of the variable simulated by MAIDEN to represent tree-growth directly with observations. However, this would imply the use of other tree-growth observations such as tree-rings density measurements, while tree-ring width represents the most widely available tree-growth observations which makes it a relevant candidate given our global scale goals. The disadvantage is that this normalization forbids us to assess error in the variance. This is why we only analyse the correlations for simplicity as using other metrics like the RMSE would not help us in this aspect.**".

*I'm confused about the use of NRCAN data in the VS-lite model. If I've understood the manuscript correctly, the NRCAN data provides daily max-min temperature and precip data. But I believe that VS-lite is designed for monthly mean data. Is NRCAN (and daily max/min values) the right data to be using for VS-lite? I'm wondering if this might contribute to the reduced skill of VS-lite.*

We agree with the reviewer that using daily maximum and minimum values could be a source of bias for VS-Lite. This problem has been highlighted in the PhD thesis of Alexandre Devers available online (https://www.theses.fr/2019GREAU029), on p.56 for example, where for France the average difference between daily average temperature and daily average temperature calculated from minimum and maximum temperature has been estimated to be around 0.5°C. The bias should be relatively weak and thus not impact so much the skill of VS-Lite.

The following information will be added in the revised manuscript on lines 166-167 (p.7), as follows:"**Note that monthly average temperature has been computed by averaging daily maximum and minimum temperature, which could lead to a small bias.**"

Alexandre Devers. Vers une réanalyse hydrométéorologique à l'échelle de la France sur les 150 dernières années par assimilation de données dans des reconstructions ensemblistes. Hydrologie. Université Grenoble Alpes, 2019. Français. NNT: 2019GREAU029. tel-02506254

*Can the authors comment on the computation cost of running MAIDEN vs VS-lite? This is particularly relevant for paleoclimate DA where an expensive PSM might be justification enough for not using it if something else is much faster.*

We agree that it is an important information to add in the manuscript. This information will be added on lines 233-236 (p.10), as follows: "**Running MAIDEN takes around 2.5 seconds on one CPU for a 50 years time span while running VS-Lite takes around 0.30 seconds. Currently, calibrating MAIDEN with our method takes around 18 hours on one CPU for a site due to the**

**high number of iterations and calibrated parameters, while the calibration method used for VS-Lite and developed by Tolwinski-Ward et al. (2013) takes only a few seconds.** ”

*p2.l51-53 This isn't actually true. Several reconstructions have assimilated proxy values directly using linear statistical "PSMs" (e.g., Hakim et al. 2016, Steiger et al. 2018, Tardif et al. 2018). While these are not physically-based, they still are a kind of PSM and the proxy values are not converted to temperature and then assimilated. Additionally there are reconstructions methods that have tested the direct assimilation of real isotope data using isotope GCMs (Steiger et al. 2017, Okazaki and Yoshimura 2019), and thus employed fully physically-based PSMs.*

We agree with the reviewer that it has not been stated clearly in our manuscript. In the introduction, we are only talking about physically-based PSMs and this will be corrected in the revised manuscript accordingly. Also, there are indeed examples where physically-based GCMs have been used with direct assimilation but for other variables (isotopes) and not for tree-rings.

We will revise the manuscript on lines 52-53 (p.2) as follows: “**However, so far, physically-based tree-rings PSMs have not been used in published reconstructions based on data assimilation using actual data.**”

*p3.l62-64 Is the inclusion of CO2 influences needed for Common Era paleoclimate though? Over most of the Common Era CO2 changes very little. Then when CO2 does start to matter, we have plentiful observations? Maybe there's some other aspect of the MAIDEN model that would be more beneficial to highlight for paleoclimatic applications? It just seems like the use of MAIDEN might not be sufficiently motivated here.*

We think that the inclusion of $CO_2$ influences is very important as models are calibrated over the recent period where $CO_2$ concentration has changed a lot. If we do not take the $CO_2$ effect into account, then it could potentially induce stationarity problems which can, ultimately, have an impact on other parameters, such as the ones related to temperature that can covariate with $CO_2$.

The following sentence will be added to the revised manuscript on lines 64-66 (p.3): “**
[revised manuscript text omitted]
 S4.**  MAIDEN calibrated parameters values (Table S3)  over the 1950-2000 period for  the twenty-one Eastern Canadian taiga sites, five aggregated Eastern Canadian taiga sites (NRCAN (5') climatic dataset, Fig. 1, Table 2) and three European sites (GHCN station data, Fig. 2, Table 2).

| Dataset | Site | GDD1 | vegphase23 | day23_flex | CanopyP | CanopyT | PercentFall | OutMax | OutLength | Cbud | h3 | st4temp | photoper | Vmax | Vb | Vip | soilb | soilip | tau |
|---|---|---|---|---|---|---|---|---|---|---|---|---|---|---|---|---|---|---|---|
| NRCAN | QC_taiga | 63.403 | 155.321 | 7.441 | 1.104 | 1.309 | 0.137 | 170.266 | 9.378 | 1.892 | 0.970 | 99.921 | 13.743 | 33.246 | -0.135 | 20.301 | -0.014 | 236.251 | 10.986 |
| NRCAN | WCORPL | 19.049 | 171.896 | 5.481 | 15.869 | 16.044 | 0.122 | 181.651 | 8.325 | 1.500 | 0.690 | 15.565 | 13.367 | 58.559 | -0.229 | 11.111 | -0.020 | 318.724 | 6.964 |
| NRCAN | WNFLRI | 74.008 | 170.197 | 8.624 | 16.625 | 17.049 | 0.099 | 177.628 | 11.309 | 1.983 | 0.324 | 42.883 | 13.348 | 26.483 | -0.136 | 10.839 | -0.023 | 260.840 | 6.672 |
| NRCAN | WL42 | 86.427 | 176.503 | 3.561 | 1.425 | 1.644 | 0.091 | 199.670 | 11.317 | 1.384 | 0.506 | 17.047 | 13.128 | 61.034 | -0.193 | 14.804 | -0.018 | 300.386 | 13.380 |
| NRCAN | WCORILE | 97.375 | 168.303 | 4.367 | 19.023 | 4.940 | 0.108 | 159.102 | 9.390 | 1.390 | 0.285 | 7.698 | 12.889 | 71.462 | -0.135 | 13.767 | -0.013 | 368.213 | 2.123 |
| NRCAN | WPOOL | 17.709 | 167.718 | 6.990 | 3.910 | 9.930 | 0.097 | 172.567 | 10.908 | 1.299 | 0.124 | 45.731 | 12.300 | 28.358 | -0.197 | 16.832 | -0.016 | 119.184 | 1.510 |
| NRCAN | WNIT | 34.580 | 165.366 | 1.194 | 14.719 | 9.647 | 0.124 | 163.836 | 11.716 | 1.082 | 0.167 | 82.783 | 13.256 | 123.684 | -0.176 | 19.069 | -0.014 | 353.154 | 9.761 |
| NRCAN | WCANE | 28.189 | 178.133 | 3.459 | 7.511 | 1.942 | 0.114 | 181.293 | 9.072 | 1.135 | 0.023 | 94.195 | 13.767 | 128.890 | -0.294 | 20.993 | -0.024 | 202.876 | 1.249 |
| NRCAN | WCEA | 102.804 | 177.515 | 7.770 | 3.976 | 8.212 | 0.107 | 186.967 | 8.398 | 1.340 | 0.076 | 43.256 | 12.936 | 125.486 | -0.163 | 23.795 | -0.018 | 374.543 | 6.827 |
| NRCAN | WDA1R | 17.256 | 159.653 | 5.514 | 15.392 | 0.170 | 0.098 | 165.542 | 10.218 | 1.488 | 0.622 | 15.382 | 13.571 | 84.079 | -0.136 | 19.056 | -0.021 | 298.486 | 3.694 |
| NRCAN | WHER | 24.016 | 180.457 | 9.596 | 10.538 | 10.287 | 0.097 | 175.104 | 10.539 | 1.794 | 0.282 | 56.515 | 13.987 | 16.204 | -0.119 | 12.972 | -0.015 | 120.703 | 4.005 |
| NRCAN | WHH1 | 34.570 | 161.348 | 6.628 | 15.612 | 18.041 | 0.103 | 178.681 | 11.085 | 1.575 | 0.253 | 22.939 | 13.779 | 26.570 | -0.133 | 18.467 | -0.023 | 114.641 | 9.420 |
| NRCAN | WHM1 | 109.541 | 177.636 | 7.965 | 15.971 | 3.851 | 0.096 | 182.295 | 9.734 | 1.711 | 0.225 | 50.227 | 12.500 | 26.902 | -0.142 | 17.533 | -0.007 | 288.438 | 2.278 |
| NRCAN | WHM2 | 39.917 | 178.020 | 2.164 | 16.710 | 18.633 | 0.127 | 172.894 | 7.970 | 1.481 | 0.479 | 32.984 | 12.304 | 63.905 | -0.145 | 14.738 | -0.022 | 398.449 | 4.107 |
| NRCAN | WL32 | 60.984 | 156.171 | 7.108 | 13.672 | 16.074 | 0.090 | 169.621 | 9.887 | 1.472 | 0.469 | 6.886 | 13.946 | 31.660 | -0.286 | 15.059 | -0.010 | 129.900 | 3.464 |
| NRCAN | WLECA | 111.354 | 167.925 | 8.528 | 2.635 | 10.267 | 0.103 | 193.841 | 11.685 | 1.234 | 0.313 | 16.742 | 12.861 | 51.242 | -0.193 | 20.333 | -0.008 | 208.170 | 2.746 |
| NRCAN | WNFL1V | 14.210 | 169.452 | 7.227 | 13.154 | 4.663 | 0.121 | 165.126 | 11.766 | 1.263 | 0.998 | 52.112 | 12.437 | 36.575 | -0.113 | 19.736 | -0.012 | 377.023 | 1.067 |
| NRCAN | WROZM | 21.270 | 167.485 | 1.875 | 18.106 | 16.159 | 0.133 | 160.417 | 4.605 | 1.151 | 0.757 | 30.170 | 13.914 | 67.561 | -0.104 | 21.151 | -0.023 | 292.789 | 2.247 |
| NRCAN | WROZX | 43.035 | 168.173 | 1.593 | 12.498 | 19.278 | 0.096 | 169.720 | 7.006 | 1.310 | 0.276 | 34.586 | 13.553 | 27.911 | -0.105 | 18.322 | -0.022 | 243.454 | 18.665 |
| NRCAN | WRT485 | 89.048 | 173.233 | 6.517 | 12.051 | 5.321 | 0.094 | 176.983 | 11.824 | 1.231 | 0.201 | 6.992 | 13.247 | 82.030 | -0.129 | 29.358 | -0.019 | 126.865 | 2.448 |
| NRCAN | WTHH | 15.963 | 167.972 | 5.069 | 0.742 | 19.062 | 0.110 | 168.375 | 11.230 | 1.049 | 0.267 | 8.399 | 13.325 | 72.763 | -0.135 | 28.040 | -0.018 | 127.278 | 2.044 |
| NRCAN | WCOR | 14.568 | 154.225 | 9.215 | 4.124 | 8.017 | 0.107 | 159.538 | 11.356 | 2.427 | 0.256 | 41.704 | 13.833 | 66.425 | -0.100 | 13.036 | -0.014 | 331.144 | 19.299 |
| NRCAN | WNFL | 96.049 | 159.796 | 3.996 | 15.297 | 9.355 | 0.092 | 184.498 | 10.207 | 2.389 | 0.265 | 47.119 | 13.405 | 41.677 | -0.177 | 12.018 | -0.020 | 273.574 | 3.263 |
| NRCAN | WDA1R_WTHH | 18.938 | 174.889 | 6.651 | 1.032 | 2.323 | 0.108 | 150.426 | 6.464 | 1.380 | 0.298 | 8.275 | 12.311 | 110.619 | -0.142 | 13.209 | -0.023 | 356.216 | 19.803 |
| NRCAN | WROZ | 26.364 | 153.589 | 1.601 | 15.775 | 18.630 | 0.147 | 161.491 | 9.386 | 1.133 | 0.620 | 2.697 | 13.254 | 96.710 | -0.124 | 14.956 | -0.014 | 396.201 | 13.193 |
| NRCAN | WH | 54.844 | 155.791 | 4.134 | 5.880 | 15.647 | 0.110 | 171.819 | 10.873 | 1.594 | 0.152 | 5.462 | 12.598 | 43.160 | -0.143 | 13.597 | -0.017 | 134.855 | 1.872 |
| GHCN | EALP | 176.466 | 92.590 | 7.835 | 3.483 | 2.651 | 0.263 | 145.638 | 3.635 | 13.142 | 0.993 | 8.250 | 14.028 | 80.946 | -0.140 | 24.093 | -0.029 | 342.517 | 14.468 |
| GHCN | SWIT179 | 43.239 | 158.136 | 2.612 | 6.654 | 7.949 | 0.500 | 224.261 | 12.397 | 9.147 | 0.467 | 15.266 | 9.822 | 54.405 | -0.175 | 14.108 | -0.056 | 304.582 | 16.755 |
| GHCN | FINL045 | 56.252 | 152.044 | 3.346 | 6.974 | 8.250 | 0.132 | 242.491 | 8.044 | 4.489 | 0.879 | 44.379 | 11.962 | 117.566 | -0.173 | 18.193 | -0.054 | 419.557 | 9.348 |

**Table S5.** MAIDEN calibrated parameters values (Table S3) over the 1950-2000 period for the twenty-one Eastern Canadian taiga sites and five aggregated Eastern Canadian taiga sites (GMF (1°) climatic dataset, Fig. 1, Table 2).

| Dataset | Site | GDD1 | vegphase23 | day23_flex | CanopyP | CanopyT | PercentFall | OutMax | OutLength | Cbud | h3 | st4temp | photoper | Vmax | Vb | Vip | soilb | soilip | tau |
|---|---|---|---|---|---|---|---|---|---|---|---|---|---|---|---|---|---|---|---|
| GMF | QC_taiga | 75.663 | 152.558 | 1.636 | 2.863 | 8.957 | 0.134 | 189.983 | 10.825 | 1.200 | 0.983 | 99.616 | 13.737 | 13.314 | -0.133 | 11.010 | -0.009 | 242.325 | 5.220 |
| GMF | WCORPL | 19.614 | 154.658 | 5.627 | 0.264 | 19.840 | 0.131 | 168.635 | 9.027 | 1.895 | 0.780 | 26.895 | 13.139 | 65.571 | -0.121 | 13.440 | -0.019 | 368.897 | 3.752 |
| GMF | WNFLR1 | 49.391 | 169.826 | 6.628 | 11.234 | 6.101 | 0.113 | 167.168 | 10.788 | 1.771 | 0.335 | 15.850 | 12.424 | 29.047 | -0.188 | 11.749 | -0.013 | 305.712 | 10.576 |
| GMF | WL42 | 58.119 | 172.639 | 7.569 | 16.189 | 7.377 | 0.109 | 179.399 | 8.269 | 1.035 | 0.535 | 1.178 | 13.411 | 18.125 | -0.176 | 10.952 | -0.011 | 297.743 | 4.432 |
| GMF | WCORILLE | 26.325 | 164.009 | 5.596 | 17.281 | 8.733 | 0.146 | 169.686 | 9.020 | 1.114 | 0.407 | 11.819 | 13.201 | 16.369 | -0.125 | 12.854 | -0.020 | 206.164 | 18.014 |
| GMF | WPOOL. | 77.772 | 173.692 | 7.052 | 2.790 | 15.853 | 0.091 | 184.851 | 11.002 | 1.336 | 0.090 | 30.447 | 13.644 | 42.324 | -0.169 | 20.228 | -0.023 | 143.943 | 10.72 |
| GMF | WNIT | 30.784 | 166.823 | 2.741 | 13.505 | 7.509 | 0.134 | 164.790 | 11.952 | 1.678 | 0.373 | 22.584 | 12.858 | 24.049 | -0.273 | 13.153 | -0.009 | 174.770 | 6.647 |
| GMF | WCANE | 70.119 | 170.101 | 3.273 | 15.928 | 8.959 | 0.144 | 185.114 | 9.475 | 1.087 | 0.105 | 27.563 | 12.575 | 137.905 | -0.280 | 18.917 | -0.018 | 381.838 | 11.640 |
| GMF | WCEA | 85.430 | 161.229 | 1.943 | 18.883 | 12.005 | 0.142 | 153.534 | 5.886 | 1.193 | 0.431 | 78.100 | 13.618 | 107.353 | -0.230 | 11.551 | -0.020 | 394.208 | 1.954 |
| GMF | WDA1R | 24.320 | 152.898 | 8.400 | 1.016 | 14.503 | 0.129 | 174.207 | 11.061 | 1.974 | 0.905 | 61.826 | 13.600 | 108.395 | -0.103 | 22.824 | -0.024 | 325.118 | 4.721 |
| GMF | WHER | 81.055 | 154.396 | 2.065 | 7.682 | 9.018 | 0.095 | 157.001 | 9.334 | 1.102 | 0.589 | 2.016 | 13.400 | 16.916 | -0.121 | 11.807 | -0.012 | 336.875 | 12.949 |
| GMF | WHH1 | 14.275 | 174.618 | 2.949 | 10.268 | 0.171 | 0.093 | 171.613 | 10.146 | 1.223 | 0.292 | 13.350 | 12.947 | 21.290 | -0.235 | 13.151 | -0.022 | 222.973 | 13.111 |
| GMF | WHM1 | 32.838 | 167.438 | 6.757 | 6.371 | 5.958 | 0.110 | 155.831 | 8.873 | 1.181 | 0.360 | 11.616 | 13.477 | 18.910 | -0.104 | 17.042 | -0.011 | 151.966 | 2.310 |
| GMF | WHM2 | 91.379 | 152.520 | 3.743 | 1.271 | 17.498 | 0.128 | 151.777 | 9.601 | 1.037 | 0.502 | 10.550 | 12.253 | 13.764 | -0.140 | 10.661 | -0.011 | 273.032 | 5.109 |
| GMF | WL32 | 95.642 | 180.031 | 4.224 | 4.193 | 14.564 | 0.098 | 177.032 | 11.805 | 1.861 | 0.073 | 34.838 | 13.185 | 53.737 | -0.203 | 20.427 | -0.025 | 143.028 | 5.336 |
| GMF | WLECA | 116.601 | 172.242 | 7.905 | 18.894 | 4.609 | 0.104 | 163.603 | 4.515 | 1.150 | 0.577 | 20.228 | 13.408 | 13.865 | -0.118 | 12.874 | -0.005 | 118.175 | 4.252 |
| GMF | WNFL1V | 58.951 | 159.826 | 8.953 | 13.153 | 12.897 | 0.114 | 178.394 | 10.591 | 1.169 | 0.480 | 8.190 | 12.041 | 70.906 | -0.106 | 29.684 | -0.014 | 246.196 | 10.527 |
| GMF | WROZM | 34.123 | 154.326 | 1.391 | 1.824 | 12.766 | 0.133 | 175.840 | 7.810 | 1.120 | 0.503 | 14.596 | 12.224 | 38.683 | -0.147 | 17.125 | -0.019 | 268.996 | 1.033 |
| GMF | WROZX | 61.982 | 157.946 | 3.203 | 7.807 | 13.029 | 0.140 | 176.400 | 8.099 | 1.644 | 0.869 | 72.129 | 13.874 | 112.519 | -0.101 | 23.261 | -0.025 | 395.291 | 5.423 |
| GMF | WRT485 | 24.015 | 169.276 | 8.934 | 18.033 | 7.811 | 0.133 | 172.579 | 11.565 | 1.588 | 0.701 | 12.111 | 12.298 | 17.253 | -0.221 | 11.145 | -0.024 | 205.932 | 3.025 |
| GMF | WTHH | 68.680 | 177.619 | 9.452 | 9.448 | 17.848 | 0.119 | 167.538 | 7.717 | 1.080 | 0.571 | 4.714 | 13.603 | 19.948 | -0.185 | 14.189 | -0.010 | 212.170 | 6.833 |
| GMF | WCOR | 20.805 | 178.574 | 2.620 | 10.231 | 19.250 | 0.105 | 154.684 | 6.307 | 1.818 | 0.469 | 12.603 | 12.393 | 42.352 | -0.117 | 12.343 | -0.009 | 376.197 | 4.200 |
| GMF | WNFL | 47.029 | 161.315 | 3.861 | 6.531 | 8.277 | 0.094 | 182.211 | 10.528 | 2.095 | 0.382 | 16.444 | 12.828 | 45.482 | -0.230 | 11.637 | -0.014 | 311.371 | 1.951 |
| GMF | WDA1R_WTHH | 39.429 | 180.587 | 4.740 | 0.007 | 10.191 | 0.120 | 167.530 | 6.059 | 1.446 | 0.568 | 3.757 | 13.472 | 26.760 | -0.124 | 11.530 | -0.009 | 290.608 | 2.714 |
| GMF | WROZ | 29.498 | 180.242 | 3.967 | 0.403 | 9.470 | 0.120 | 175.502 | 8.817 | 1.364 | 0.338 | 8.931 | 13.680 | 44.734 | -0.103 | 11.303 | -0.009 | 388.244 | 12.917 |
| GMF | WH | 66.488 | 154.640 | 2.067 | 9.850 | 1.462 | 0.098 | 182.743 | 10.232 | 2.314 | 0.260 | 9.865 | 13.106 | 33.213 | -0.103 | 10.300 | -0.013 | 225.312 | 4.833 |

**Table S6.** MAIDEN calibrated parameters values (Table S3) over the 1950-2000 period for the twenty-one Eastern Canadian taiga sites and five aggregated Eastern Canadian taiga sites (20CRv2c corr. (2°) climatic dataset, Fig. 1, Table 2).

| Dataset | Site | GDD1 | vegphase23 | day23_flex | CanopyP | CanopyT | PercentFall | OutMax | OutLength | Cbud | h3 | si4temp | photoper | Vmax | Vb | Vip | soilb | soilip | tau |
|---|---|---|---|---|---|---|---|---|---|---|---|---|---|---|---|---|---|---|---|
| 20CRv2c corr. | QC_taiga | 113.370 | 152.798 | 8.744 | 0.222 | 16.280 | 0.147 | 171.699 | 7.445 | 1.342 | 0.998 | 90.354 | 13.635 | 27.746 | -0.102 | 12.668 | -0.006 | 142.220 | 11.427 |
| 20CRv2c corr. | WCORPL | 67.994 | 167.436 | 7.342 | 11.186 | 15.686 | 0.111 | 167.320 | 11.872 | 1.700 | 0.129 | 87.342 | 12.581 | 32.381 | -0.109 | 21.452 | -0.010 | 214.488 | 17.015 |
| 20CRv2c corr. | WNFLR1 | 78.261 | 174.734 | 6.191 | 2.768 | 8.093 | 0.092 | 162.590 | 5.851 | 1.877 | 0.192 | 58.798 | 13.299 | 65.388 | -0.122 | 15.216 | -0.023 | 287.940 | 2.384 |
| 20CRv2c corr. | WL42 | 29.260 | 178.089 | 9.434 | 13.527 | 17.155 | 0.093 | 167.099 | 11.168 | 1.024 | 0.954 | 4.132 | 13.363 | 30.286 | -0.113 | 11.358 | -0.022 | 357.574 | 19.205 |
| 20CRv2c corr. | WCORILE | 12.599 | 174.353 | 6.890 | 0.700 | 4.433 | 0.129 | 195.443 | 9.126 | 1.850 | 0.930 | 85.955 | 13.050 | 53.561 | -0.296 | 10.832 | -0.025 | 365.462 | 6.040 |
| 20CRv2c corr. | WPOOL | 112.458 | 180.716 | 8.072 | 14.904 | 17.606 | 0.097 | 198.678 | 11.974 | 1.244 | 0.120 | 25.553 | 13.383 | 78.202 | -0.130 | 29.083 | -0.017 | 182.063 | 4.687 |
| 20CRv2c corr. | WNIT | 19.219 | 176.941 | 6.150 | 12.806 | 9.017 | 0.136 | 151.973 | 11.449 | 2.039 | 0.427 | 35.933 | 13.733 | 134.551 | -0.269 | 16.502 | -0.024 | 390.727 | 9.147 |
| 20CRv2c corr. | WCANE | 78.482 | 167.604 | 9.756 | 10.507 | 8.310 | 0.099 | 164.636 | 11.268 | 2.186 | 0.994 | 90.511 | 12.379 | 51.995 | -0.289 | 16.932 | -0.019 | 236.149 | 13.742 |
| 20CRv2c corr. | WCEA | 81.167 | 178.414 | 7.441 | 0.210 | 15.923 | 0.116 | 169.741 | 4.922 | 1.348 | 0.389 | 75.341 | 13.536 | 58.210 | -0.127 | 23.997 | -0.012 | 382.036 | 5.418 |
| 20CRv2c corr. | WDA1R | 104.308 | 160.431 | 3.691 | 5.556 | 7.740 | 0.140 | 161.868 | 10.006 | 1.481 | 0.659 | 1.507 | 13.440 | 104.597 | -0.103 | 21.727 | -0.023 | 344.532 | 6.344 |
| 20CRv2c corr. | WHER | 63.043 | 166.470 | 2.476 | 17.934 | 16.551 | 0.091 | 177.202 | 10.553 | 1.785 | 0.195 | 61.706 | 13.385 | 15.438 | -0.113 | 10.067 | -0.021 | 314.001 | 14.469 |
| 20CRv2c corr. | WHH1 | 89.238 | 162.196 | 2.490 | 14.260 | 5.373 | 0.113 | 184.458 | 10.997 | 1.528 | 0.299 | 16.712 | 13.163 | 35.179 | -0.100 | 24.270 | -0.022 | 257.149 | 1.324 |
| 20CRv2c corr. | WHM1 | 89.658 | 165.179 | 2.332 | 15.537 | 19.911 | 0.097 | 174.207 | 11.409 | 1.661 | 0.047 | 98.363 | 12.489 | 124.366 | -0.154 | 29.038 | -0.012 | 233.181 | 3.418 |
| 20CRv2c corr. | WHM2 | 110.167 | 170.088 | 4.846 | 0.043 | 12.045 | 0.105 | 165.444 | 7.855 | 1.318 | 0.274 | 34.803 | 12.197 | 19.156 | -0.111 | 16.353 | -0.023 | 167.282 | 7.595 |
| 20CRv2c corr. | WL32 | 116.547 | 178.676 | 5.965 | 5.265 | 15.111 | 0.092 | 184.089 | 10.907 | 1.766 | 0.053 | 58.663 | 13.483 | 42.173 | -0.269 | 16.806 | -0.014 | 145.674 | 1.660 |
| 20CRv2c corr. | WLECA | 90.354 | 180.902 | 5.626 | 11.212 | 8.273 | 0.109 | 199.300 | 7.751 | 1.013 | 0.010 | 49.618 | 12.033 | 22.119 | -0.212 | 13.036 | -0.015 | 313.818 | 12.315 |
| 20CRv2c corr. | WNFL1V | 40.318 | 179.836 | 6.997 | 1.668 | 9.648 | 0.129 | 171.170 | 8.712 | 1.515 | 0.137 | 57.226 | 12.209 | 30.784 | -0.127 | 20.251 | -0.012 | 171.664 | 9.904 |
| 20CRv2c corr. | WROZM | 63.805 | 164.546 | 2.513 | 2.971 | 13.699 | 0.101 | 169.451 | 11.196 | 1.358 | 0.154 | 70.833 | 12.796 | 15.280 | -0.103 | 11.943 | -0.017 | 153.426 | 18.963 |
| 20CRv2c corr. | WROZX | 11.256 | 158.783 | 1.475 | 6.717 | 11.490 | 0.100 | 169.016 | 8.972 | 1.305 | 0.162 | 68.989 | 13.289 | 15.439 | -0.129 | 12.107 | -0.022 | 197.314 | 15.718 |
| 20CRv2c corr. | WRT485 | 102.122 | 173.485 | 9.331 | 3.189 | 13.364 | 0.121 | 181.631 | 11.174 | 1.024 | 0.318 | 3.755 | 12.111 | 67.786 | -0.127 | 27.347 | -0.005 | 366.987 | 3.936 |
| 20CRv2c corr. | WTHH | 48.844 | 171.447 | 6.643 | 0.289 | 9.060 | 0.139 | 170.923 | 8.826 | 1.182 | 0.534 | 7.378 | 12.208 | 21.961 | -0.136 | 16.102 | -0.010 | 143.590 | 3.259 |
| 20CRv2c corr. | WCOR | 56.468 | 179.615 | 3.574 | 16.008 | 1.242 | 0.114 | 157.886 | 7.560 | 2.017 | 0.263 | 14.179 | 13.968 | 102.747 | -0.115 | 16.294 | -0.019 | 397.049 | 10.996 |
| 20CRv2c corr. | WNFL | 20.771 | 164.646 | 4.922 | 5.208 | 12.450 | 0.127 | 155.527 | 11.352 | 2.304 | 0.078 | 63.865 | 12.954 | 39.707 | -0.116 | 17.774 | -0.016 | 211.665 | 3.658 |
| 20CRv2c corr. | WDA1R_WTHH | 59.728 | 175.202 | 8.912 | 7.291 | 15.136 | 0.110 | 168.798 | 10.731 | 1.586 | 0.169 | 25.445 | 12.855 | 72.407 | -0.127 | 13.938 | -0.019 | 382.916 | 3.795 |
| 20CRv2c corr. | WROZ | 19.524 | 173.178 | 3.238 | 19.948 | 17.295 | 0.091 | 151.863 | 8.737 | 1.837 | 0.247 | 61.332 | 12.374 | 89.821 | -0.107 | 14.736 | -0.021 | 397.978 | 15.062 |
| 20CRv2c corr. | WH | 119.083 | 165.796 | 1.176 | 2.114 | 8.057 | 0.104 | 184.843 | 11.733 | 2.771 | 0.067 | 18.052 | 13.788 | 76.144 | -0.105 | 22.057 | -0.006 | 328.383 | 1.022 |

**Table S7.** MAIDEN calibrated parameters values (Table S3) over the 1950-2000 period for the twenty-one Eastern Canadian taiga sites and five aggregated Eastern Canadian taiga sites (20CRv2c (2°) climatic dataset, Fig. 1, Table 2).

| Dataset | Site | GDD1 | vegphase23 | day23_flex | CanopyP | CanopyT | PercentFall | OuMax | OutLength | Cbud | h3 | st4temp | photoper | Vmax | Vb | Vip | soilb | soiiip | tau |
|---|---|---|---|---|---|---|---|---|---|---|---|---|---|---|---|---|---|---|---|
| 20CRv2c | QC_taiga | 75.720 | 162.168 | 9.429 | 3.806 | 4.436 | 0.142 | 161.784 | 7.128 | 2.868 | 0.947 | 95.218 | 13.485 | 22.246 | -0.157 | 13.200 | -0.008 | 397.024 | 2.846 |
| 20CRv2c | WCORPL | 44.431 | 158.330 | 7.142 | 2.826 | 19.045 | 0.099 | 187.010 | 5.206 | 1.043 | 0.026 | 36.677 | 13.618 | 103.760 | -0.116 | 20.246 | -0.018 | 374.728 | 19.391 |
| 20CRv2c | WNFLR1 | 75.809 | 167.038 | 1.718 | 13.416 | 9.042 | 0.111 | 179.207 | 11.314 | 1.385 | 0.099 | 84.442 | 13.892 | 39.007 | -0.135 | 18.911 | -0.006 | 369.090 | 2.293 |
| 20CRv2c | WL42 | 83.970 | 161.752 | 3.598 | 18.021 | 15.015 | 0.117 | 196.304 | 10.705 | 1.179 | 0.471 | 89.359 | 12.869 | 21.043 | -0.193 | 11.375 | -0.022 | 248.490 | 12.055 |
| 20CRv2c | WCORILE | 101.474 | 154.897 | 6.015 | 7.926 | 5.971 | 0.099 | 167.905 | 11.464 | 1.061 | 0.057 | 35.086 | 12.627 | 68.195 | -0.152 | 14.843 | -0.017 | 397.308 | 11.180 |
| 20CRv2c | WPOOL | 107.070 | 163.868 | 5.215 | 12.717 | 14.609 | 0.097 | 181.672 | 11.563 | 1.112 | 0.119 | 38.952 | 12.915 | 20.150 | -0.206 | 12.698 | -0.015 | 181.127 | 12.122 |
| 20CRv2c | WNIT | 46.340 | 174.983 | 5.523 | 0.065 | 19.960 | 0.135 | 170.592 | 11.654 | 1.573 | 0.321 | 26.871 | 13.769 | 16.621 | -0.275 | 10.872 | -0.010 | 130.537 | 18.718 |
| 20CRv2c | WCANE | 117.593 | 165.327 | 6.518 | 9.262 | 8.489 | 0.095 | 190.972 | 5.377 | 1.301 | 1.000 | 55.147 | 13.234 | 24.946 | -0.191 | 15.154 | -0.018 | 212.148 | 15.867 |
| 20CRv2c | WCEA | 16.999 | 178.818 | 2.874 | 3.271 | 4.543 | 0.104 | 168.800 | 7.396 | 1.062 | 0.250 | 85.997 | 12.675 | 19.907 | -0.270 | 10.342 | -0.020 | 358.779 | 9.664 |
| 20CRv2c | WDA1R | 44.501 | 175.164 | 5.003 | 11.839 | 13.155 | 0.108 | 170.377 | 9.474 | 1.092 | 0.243 | 81.989 | 12.506 | 84.785 | -0.112 | 17.687 | -0.022 | 348.379 | 1.489 |
| 20CRv2c | WHER | 55.843 | 154.795 | 8.234 | 8.967 | 17.999 | 0.097 | 180.100 | 11.782 | 1.427 | 0.310 | 54.276 | 12.896 | 16.450 | -0.101 | 13.173 | -0.017 | 202.653 | 15.885 |
| 20CRv2c | WHH1 | 115.526 | 170.285 | 2.370 | 12.582 | 17.775 | 0.119 | 189.149 | 10.913 | 1.159 | 0.309 | 17.661 | 13.287 | 38.614 | -0.139 | 21.284 | -0.020 | 211.300 | 3.248 |
| 20CRv2c | WHM1 | 66.111 | 172.640 | 8.305 | 0.824 | 4.900 | 0.091 | 193.059 | 11.798 | 1.214 | 0.164 | 18.974 | 13.369 | 28.720 | -0.181 | 16.652 | -0.008 | 287.187 | 2.558 |
| 20CRv2c | WHM2 | 116.989 | 170.119 | 8.727 | 6.623 | 1.158 | 0.150 | 194.261 | 11.175 | 1.263 | 0.389 | 59.636 | 12.473 | 29.252 | -0.129 | 18.507 | -0.011 | 169.127 | 1.643 |
| 20CRv2c | WL32 | 15.396 | 176.894 | 2.706 | 6.839 | 15.526 | 0.122 | 170.542 | 11.614 | 1.120 | 0.182 | 87.455 | 13.711 | 110.276 | -0.295 | 14.399 | -0.022 | 353.621 | 1.233 |
| 20CRv2c | WLECA | 100.826 | 174.227 | 3.438 | 17.519 | 17.296 | 0.096 | 186.546 | 10.267 | 1.244 | 0.302 | 37.580 | 12.352 | 27.111 | -0.132 | 18.973 | -0.023 | 179.091 | 1.122 |
| 20CRv2c | WNFLIV | 79.176 | 173.450 | 1.076 | 16.285 | 11.663 | 0.129 | 158.549 | 7.783 | 1.072 | 0.293 | 85.854 | 12.496 | 57.088 | -0.263 | 11.029 | -0.021 | 338.566 | 2.186 |
| 20CRv2c | WROZM | 83.289 | 168.105 | 1.265 | 14.963 | 12.219 | 0.094 | 166.930 | 10.533 | 1.808 | 0.291 | 40.019 | 12.166 | 19.213 | -0.115 | 10.790 | -0.006 | 306.374 | 3.024 |
| 20CRv2c | WROZX | 24.634 | 167.104 | 2.377 | 4.668 | 10.024 | 0.125 | 174.838 | 11.551 | 1.205 | 0.147 | 93.026 | 13.551 | 47.811 | -0.106 | 26.302 | -0.013 | 183.486 | 19.587 |
| 20CRv2c | WRT485 | 103.114 | 171.036 | 1.519 | 11.590 | 19.660 | 0.098 | 185.015 | 11.226 | 1.171 | 0.134 | 19.026 | 12.543 | 126.788 | -0.159 | 28.423 | -0.014 | 238.127 | 1.085 |
| 20CRv2c | WTHH | 45.979 | 154.621 | 7.085 | 13.906 | 8.149 | 0.117 | 195.802 | 11.858 | 1.393 | 0.700 | 63.045 | 13.618 | 41.220 | -0.106 | 25.134 | -0.006 | 127.995 | 6.676 |
| 20CRv2c | WCOR | 89.792 | 169.235 | 3.771 | 8.555 | 10.056 | 0.110 | 190.627 | 10.764 | 1.682 | 0.433 | 10.341 | 13.024 | 119.892 | -0.145 | 13.926 | -0.019 | 357.625 | 16.816 |
| 20CRv2c | WNFL | 81.661 | 173.171 | 2.122 | 18.413 | 9.513 | 0.102 | 180.427 | 11.517 | 1.867 | 0.064 | 57.458 | 13.358 | 36.924 | -0.103 | 15.964 | -0.006 | 292.632 | 14.260 |
| 20CRv2c | WDA1R_WTHH | 118.615 | 156.106 | 2.599 | 0.986 | 19.844 | 0.101 | 176.546 | 11.917 | 1.675 | 0.301 | 14.910 | 12.911 | 100.050 | -0.109 | 15.932 | -0.015 | 372.220 | 19.135 |
| 20CRv2c | WROZ | 81.338 | 152.814 | 7.145 | 0.332 | 5.040 | 0.117 | 186.358 | 11.717 | 1.722 | 0.214 | 53.164 | 12.543 | -0.122 | 10.222 | -0.009 | 394.353 | 15.502 | 1.771 |
| 20CRv2c | WH | 97.027 | 168.575 | 6.833 | 17.466 | 8.474 | 0.092 | 187.697 | 11.841 | 2.264 | 0.032 | 70.550 | 13.202 | 57.902 | -0.106 | 17.880 | -0.005 | 118.539 | 1.771 |

**Table S8.** VS-Lite calibrated parameters values (Sect. 2.3.2) over the 1950-2000 period for the twenty-one Eastern Canadian taiga sites (NRCAN (5') climatic dataset, Fig. 1a, Table 2) and three European sites (GHCN station data, Fig. 2, Table 2).

| Site Dataset | Time period Sites | Station name T1 | Station lat/lon T2 | Station elevation M1 | M2 |
|---|---|---|---|---|---|
| FINL NRCAN | 1900-1944/1950-2000 QC_taiga | Sodankyla 2.430 | 67.37N26.65E 15.727 | 179m 0.053 | 0.429 |
| EALP NRCAN | 1950-2000 WCORPL | Zugspitze 4.612 | 47.42N10.99E 12.497 | 2964m 0.035 | 0.275 |
| NRCAN | 1910-1949 WNFLR1 | Innsbruck 4.914 | 47.27N11.4E 11.493 | 577m 0.033 | 0.357 |
| NRCAN | WL42 | 7.259 | 11.658 | 0.070 | 0.457 |
| NRCAN | WCORILE | 3.058 | 12.002 | 0.032 | 0.194 |
| NRCAN | WPOOL | 7.899 | 11.514 | 0.066 | 0.194 |
| NRCAN | WNIT | 7.876 | 12.118 | 0.016 | 0.230 |
| NRCAN | WCANE | 7.264 | 11.557 | 0.077 | 0.171 |
| NRCAN | WCEA | 5.745 | 12.363 | 0.074 | 0.443 |
| NRCAN | WDA1R | 1.316 | 14.399 | 0.053 | 0.183 |
| NRCAN | WHER | 2.795 | 19.393 | 0.058 | 0.258 |
| NRCAN | WHH1 | 7.490 | 11.677 | 0.007 | 0.190 |
| NRCAN | WHM1 | 7.660 | 12.939 | 0.017 | 0.220 |
| NRCAN | WHM2 | 8.843 | 12.165 | 0.040 | 0.168 |
| NRCAN | WL32 | 7.642 | 13.785 | 0.013 | 0.231 |
| NRCAN | WLECA | 8.389 | 12.148 | 0.032 | 0.169 |
| NRCAN | WNFL1V | 3.575 | 11.542 | 0.086 | 0.465 |
| NRCAN | WROZM | 1.726 | 11.656 | 0.027 | 0.153 |
| NRCAN | WROZX | 6.170 | 11.382 | 0.070 | 0.473 |
| NRCAN | WRT485 | 2.014 | 17.012 | 0.001 | 0.158 |
| NRCAN | WTHH | 3.996 | 13.065 | 0.020 | 0.119 |
| GHCN | EALP | 8.242 | 22.117 | 0.058 | 0.277 |
| GHCN | SWIT179 | 1910-2000 1.480 | Saentis 21.912 | 47.25N9.35E 0.052 | 2502m 0.294 |
| GHCN | FINL045 | 2.517 | 19.159 | 0.007 | 0.120 |

**Table S9.** VS-Lite calibrated parameters values (Sect. 2.3.2) over the 1950-2000 period for the twenty-one Eastern Canadian taiga sites (GMF (1°) climatic dataset, Fig. 1a, Table 2).

| Dataset | Sites | T1 | T2 | M1 | M2 |
|---------|---------|-------|--------|-------|-------|
| GMF | QC_taiga | 7.934 | 20.259 | 0.036 | 0.210 |
| GMF | WCORPL | 2.574 | 12.366 | 0.027 | 0.233 |
| GMF | WNFLR1 | 3.124 | 10.795 | 0.018 | 0.404 |
| GMF | WL42 | 6.973 | 10.861 | 0.036 | 0.378 |
| GMF | WCORILE | 2.585 | 12.279 | 0.025 | 0.132 |
| GMF | WPOOL | 8.036 | 11.556 | 0.042 | 0.266 |
| GMF | WNIT | 8.193 | 13.365 | 0.028 | 0.219 |
| GMF | WCANE | 7.517 | 12.862 | 0.089 | 0.482 |
| GMF | WCEA | 6.072 | 11.476 | 0.080 | 0.469 |
| GMF | WDA1R | 1.613 | 22.429 | 0.003 | 0.318 |
| GMF | WHER | 4.808 | 12.558 | 0.040 | 0.439 |
| GMF | WHH1 | 7.303 | 11.754 | 0.061 | 0.259 |
| GMF | WHM1 | 2.750 | 13.427 | 0.009 | 0.223 |
| GMF | WHM2 | 5.479 | 12.363 | 0.023 | 0.185 |
| GMF | WL32 | 8.300 | 15.367 | 0.007 | 0.355 |
| GMF | WLECA | 7.638 | 11.770 | 0.017 | 0.464 |
| GMF | WNFL1V | 3.241 | 11.483 | 0.080 | 0.468 |
| GMF | WROZM | 1.867 | 15.193 | 0.060 | 0.386 |
| GMF | WROZX | 1.470 | 14.070 | 0.055 | 0.154 |
| GMF | WRT485 | 1.141 | 17.046 | 0.075 | 0.386 |
| GMF | WTHH | 3.033 | 13.675 | 0.012 | 0.138 |

**Table S10.** VS-Lite calibrated parameters values (Sect. 2.3.2) over the 1950-2000 period for the twenty-one Eastern Canadian taiga sites (20CRv2c corr. (2°) climatic dataset, Fig. 1a, Table 2).

| Dataset | Sites | T1 | T2 | M1 | M2 |
|---|---|---|---|---|---|
| 20CRv2c corr. | QC_taiga | 7.000 | 14.214 | 0.094 | 0.436 |
| 20CRv2c corr. | WCORPL | 1.996 | 11.968 | 0.043 | 0.276 |
| 20CRv2c corr. | WNFLR1 | 2.443 | 19.159 | 0.011 | 0.246 |
| 20CRv2c corr. | WL42 | 7.672 | 11.259 | 0.080 | 0.447 |
| 20CRv2c corr. | WCORILE | 3.102 | 12.325 | 0.056 | 0.254 |
| 20CRv2c corr. | WPOOL | 6.812 | 10.631 | 0.005 | 0.221 |
| 20CRv2c corr. | WNIT | 8.347 | 12.275 | 0.055 | 0.201 |
| 20CRv2c corr. | WCANE | 8.277 | 12.194 | 0.017 | 0.200 |
| 20CRv2c corr. | WCEA | 2.681 | 12.493 | 0.043 | 0.410 |
| 20CRv2c corr. | WDA1R | 3.382 | 18.603 | 0.013 | 0.295 |
| 20CRv2c corr. | WHER | 4.768 | 12.783 | 0.027 | 0.196 |
| 20CRv2c corr. | WHH1 | 7.464 | 11.322 | 0.058 | 0.116 |
| 20CRv2c corr. | WHM1 | 8.472 | 15.277 | 0.082 | 0.258 |
| 20CRv2c corr. | WHM2 | 8.383 | 18.934 | 0.053 | 0.218 |
| 20CRv2c corr. | WL32 | 8.446 | 14.245 | 0.011 | 0.108 |
| 20CRv2c corr. | WLECA | 7.556 | 13.389 | 0.023 | 0.446 |
| 20CRv2c corr. | WNFL1V | 3.803 | 15.342 | 0.011 | 0.168 |
| 20CRv2c corr. | WROZM | 8.262 | 14.324 | 0.001 | 0.256 |
| 20CRv2c corr. | WROZX | 8.633 | 14.984 | 0.017 | 0.262 |
| 20CRv2c corr. | WRT485 | 8.381 | 15.478 | 0.016 | 0.189 |
| 20CRv2c corr. | WTHH | 3.802 | 15.778 | 0.033 | 0.105 |

**Table S11.** VS-Lite calibrated parameters values (Sect. 2.3.2) over the 1950-2000 period for the twenty-one Eastern Canadian taiga sites (20CRv2c (2°) climatic dataset, Fig. 1a, Table 2).

| Dataset | Sites | T1 | T2 | M1 | M2 |
|---------|-------|-----|-----|-----|-----|
| 20CRv2c | QC_taiga | 8.378 | 13.382 | 0.036 | 0.319 |
| 20CRv2c | WCORPL | 7.532 | 18.410 | 0.036 | 0.270 |
| 20CRv2c | WNFLR1 | 8.399 | 19.795 | 0.014 | 0.110 |
| 20CRv2c | WL42 | 6.012 | 10.591 | 0.031 | 0.314 |
| 20CRv2c | WCORILE | 7.629 | 10.677 | 0.047 | 0.262 |
| 20CRv2c | WPOOL | 7.219 | 10.537 | 0.076 | 0.281 |
| 20CRv2c | WNIT | 7.990 | 12.538 | 0.035 | 0.267 |
| 20CRv2c | WCANE | 7.118 | 10.445 | 0.015 | 0.279 |
| 20CRv2c | WCEA | 5.313 | 15.658 | 0.019 | 0.238 |
| 20CRv2c | WDA1R | 8.167 | 19.349 | 0.088 | 0.194 |
| 20CRv2c | WHER | 3.440 | 17.681 | 0.062 | 0.366 |
| 20CRv2c | WHH1 | 6.951 | 19.205 | 0.051 | 0.366 |
| 20CRv2c | WHM1 | 7.395 | 22.139 | 0.031 | 0.266 |
| 20CRv2c | WHM2 | 7.551 | 18.823 | 0.024 | 0.212 |
| 20CRv2c | WL32 | 8.308 | 14.045 | 0.008 | 0.234 |
| 20CRv2c | WLECA | 6.798 | 14.509 | 0.050 | 0.391 |
| 20CRv2c | WNFL1V | 8.604 | 15.787 | 0.042 | 0.153 |
| 20CRv2c | WROZM | 8.131 | 12.693 | 0.060 | 0.133 |
| 20CRv2c | WROZX | 8.645 | 16.846 | 0.035 | 0.205 |
| 20CRv2c | WRT485 | 7.555 | 20.034 | 0.019 | 0.210 |
| 20CRv2c | WTHH | 6.906 | 20.691 | 0.014 | 0.240 |

**Carbon allocation parameters for QC_taiga**

[Figure]

**Figure S1.** Posterior frequency distributions of carbon allocation parameters (Table S3) at QC_taiga site (NRCAN (5') climatic dataset) (Fig. 1a, Table 2) for the 1950-2000 calibration period.

**Photosynthesis parameters for QC_taiga**

[Figure]

[Figure]

[Figure]

[Figure]

[Figure]

[Figure]

**Figure S2.** Posterior frequency distributions of photosynthesis parameters (Table S3) at QC_taiga site (NRCAN (5') climatic dataset) (Fig. 1a, Table 2) for the 1950-2000 calibration period.

**Carbon allocation parameters for WCORPL**

[Figure]

**Figure S3.** As in Fig. S1 at WCORPL site.

**Photosynthesis parameters for WCORPL**

[Figure]

[Figure]

[Figure]

[Figure]

[Figure]

[Figure]

**Figure S4.** As in Fig. S2 at WCORPL site.

**Carbon allocation parameters for WCANE**

[Figure]

**Figure S5.** As in Fig. S1 at WCANE site.

**Photosynthesis parameters for WCANE**

[Figure]

[Figure]

[Figure]

[Figure]

[Figure]

[Figure]

**Figure S6.** As in Fig. S2 at WCANE site.

**Carbon allocation parameters for WCEA**

[Figure]

**Figure S7.** As in Fig. S1 at WCEA site.

**Photosynthesis parameters for WCEA**

[Figure]

[Figure]

[Figure]

[Figure]

[Figure]

[Figure]

**Figure S8.** As in Fig. S2 at WCEA site.

**Carbon allocation parameters for WCORILE**

[Figure]

**Figure S9.** As in Fig. S1 at WCORILE site.

**Photosynthesis parameters for WCORILE**

[Figure]

[Figure]

[Figure]

[Figure]

[Figure]

[Figure]

**Figure S10.** As in Fig. S2 at WCORILE site.

**Carbon allocation parameters for WDA1R**

[Figure]

**Figure S11.** As in Fig. S1 at WDA1R site.

**Photosynthesis parameters for WDA1R**

[Figure]

[Figure]

[Figure]

[Figure]

[Figure]

[Figure]

**Figure S12.** As in Fig. S2 at WDA1R site.

**Carbon allocation parameters for WHER**

[Figure]

**Figure S13.** As in Fig. S1 at WHER site.

**Photosynthesis parameters for WHER**

[Figure]

[Figure]

[Figure]

[Figure]

[Figure]

[Figure]

**Figure S14.** As in Fig. S2 at WHER site.

**Carbon allocation parameters for WHH1**

[Figure]

**Figure S15.** As in Fig. S1 at WHH1 site.

**Photosynthesis parameters for WHH1**

[Figure]

[Figure]

[Figure]

[Figure]

[Figure]

[Figure]

**Figure S16.** As in Fig. S2 at WHH1 site.

**Carbon allocation parameters for WHM1**

[Figure]

**Figure S17.** As in Fig. S1 at WHM1 site.

**Photosynthesis parameters for WHM1**

[Figure]

[Figure]

[Figure]

[Figure]

[Figure]

[Figure]

**Figure S18.** As in Fig. S2 at WHM1 site.

**Carbon allocation parameters for WHM2**

[Figure]

**Figure S19.** As in Fig. S1 at WHM2 site.

**Photosynthesis parameters for WHM2**

[Figure]

[Figure]

[Figure]

[Figure]

[Figure]

[Figure]

**Figure S20.** As in Fig. S2 at WHM2 site.

**Carbon allocation parameters for WL32**

[Figure]

**Figure S21.** As in Fig. S1 at WL32 site.

**Photosynthesis parameters for WL32**

[Figure]

[Figure]

[Figure]

[Figure]

[Figure]

[Figure]

**Figure S22.** As in Fig. S2 at WL32 site.

**Carbon allocation parameters for WL42**

[Figure]

**Figure S23.** As in Fig. S1 at WL42 site.

**Photosynthesis parameters for WL42**

[Figure]

[Figure]

[Figure]

[Figure]

[Figure]

[Figure]

**Figure S24.** As in Fig. S2 at WL42 site.

**Carbon allocation parameters for WLECA**

[Figure]

**Figure S25.** As in Fig. S1 at WLECA site.

**Photosynthesis parameters for WLECA**

[Figure]

[Figure]

[Figure]

[Figure]

[Figure]

[Figure]

**Figure S26.** As in Fig. S2 at WLECA site.

**Carbon allocation parameters for WNFL1V**

[Figure]

**Figure S27.** As in Fig. S1 at WNFL1V site.

**Photosynthesis parameters for WNFL1V**

[Figure]

[Figure]

[Figure]

[Figure]

[Figure]

[Figure]

**Figure S28.** As in Fig. S2 at WNFL1V site.

**Carbon allocation parameters for WNFLR1**

[Figure]

**Figure S29.** As in Fig. S1 at WNFLR1 site.

**Photosynthesis parameters for WNFLR1**

[Figure]

[Figure]

[Figure]

[Figure]

[Figure]

[Figure]

**Figure S30.** As in Fig. S2 at WNFLR1 site.

**Carbon allocation parameters for WNIT**

[Figure]

**Figure S31.** As in Fig. S1 at WNIT site.

**Photosynthesis parameters for WNIT**

[Figure]

[Figure]

[Figure]

[Figure]

[Figure]

[Figure]

**Figure S32.** As in Fig. S2 at WNIT site.

**Carbon allocation parameters for WPOOL**

[Figure]

**Figure S33.** As in Fig. S1 at WPOOL site.

**Photosynthesis parameters for WPOOL**

[Figure]

[Figure]

[Figure]

[Figure]

[Figure]

[Figure]

**Figure S34.** As in Fig. S2 at WPOOL site.

**Carbon allocation parameters for WROZM**

[Figure]

**Figure S35.** As in Fig. S1 at WROZM site.

**Photosynthesis parameters for WROZM**

[Figure]

[Figure]

[Figure]

[Figure]

[Figure]

[Figure]

**Figure S36.** As in Fig. S2 at WROZM site.

**Carbon allocation parameters for WROZX**

[Figure]

**Figure S37.** As in Fig. S1 at WROZX site.

**Photosynthesis parameters for WROZX**

[Figure]

[Figure]

[Figure]

[Figure]

[Figure]

[Figure]

**Figure S38.** As in Fig. S2 at WROZX site.

**Carbon allocation parameters for WRT485**

[Figure]

**Figure S39.** As in Fig. S1 at WRT485 site.

**Photosynthesis parameters for WRT485**

[Figure]

[Figure]

[Figure]

[Figure]

[Figure]

[Figure]

**Figure S40.** As in Fig. S2 at WRT485 site.

**Carbon allocation parameters for WTHH**

[Figure]

**Figure S41.** As in Fig. S1 at WTHH site.

**Photosynthesis parameters for WTHH**

[Figure]

[Figure]

[Figure]

[Figure]

[Figure]

[Figure]

**Figure S42.** As in Fig. S2 at WTHH site.

**Carbon allocation parameters for WROZ**

[Figure]

**Figure S43.** Posterior frequency distributions of carbon allocation parameters (Table S3) at WROZ site (NRCAN (5') climatic dataset) (Fig. 1b, Table 2) for the 1950-2000 calibration period.

**Photosynthesis parameters for WROZ**

[Figure]

[Figure]

[Figure]

[Figure]

[Figure]

[Figure]

**Figure S44.** Posterior frequency distributions of photosynthesis parameters (Table S3) at WROZ site (NRCAN (5') climatic dataset) (Fig. 1b, Table 2) for the 1950-2000 calibration period.

**Carbon allocation parameters for WH**

[Figure]

**Figure S45.** As in Fig. S43 at WH site.

**Photosynthesis parameters for WH**

[Figure]

[Figure]

[Figure]

[Figure]

[Figure]

[Figure]

**Figure S46.** As in Fig. S44 at WH site.

**Carbon allocation parameters for WNFL**

[Figure]

**Figure S47.** As in Fig. S43 at WNFL site.

**Photosynthesis parameters for WNFL**

[Figure]

[Figure]

[Figure]

[Figure]

[Figure]

[Figure]

**Figure S48.** As in Fig. S44 at WNFL site.

**Carbon allocation parameters for WCOR**

[Figure]

**Figure S49.** As in Fig. S43 at WCOR site.

**Photosynthesis parameters for WCOR**

[Figure]

[Figure]

[Figure]

[Figure]

[Figure]

[Figure]

**Figure S50.** As in Fig. S44 at WCOR site.

**Carbon allocation parameters for WDA1R_WTHH**

[Figure]

**Figure S51.** As in Fig. S43 at WDA1R_WTHH site.

**Photosynthesis parameters for WDA1R_WTHH**

[Figure]

[Figure]

[Figure]

[Figure]

[Figure]

[Figure]

**Figure S52.** As in Fig. S44 at WDA1R_WTHH site.

**Carbon allocation parameters for EALP**

[Figure]

**Figure S53.** Posterior frequency distributions of carbon allocation parameters (Table S3) at EALP site (GHCN climate data) (Fig. 2, Table 2) for the 1950-2000 calibration period.

**Photosynthesis parameters for EALP**

[Figure]

[Figure]

[Figure]

[Figure]

[Figure]

[Figure]

**Figure S54.** Posterior frequency distributions of photosynthesis parameters (Table S3) at EALP site (GHCN climate data) (Fig. 2, Table 2) for the 1950-2000 calibration period.

**Carbon allocation parameters for SWIT179**

[Figure]

**Figure S55.** As in Fig. S53 at SWIT179 site.

**Photosynthesis parameters for SWIT179**

[Figure]

[Figure]

[Figure]

[Figure]

[Figure]

[Figure]

**Figure S56.** As in Fig. S54 at SWIT179 site.

**Carbon allocation parameters for FINL045**

[Figure]

**Figure S57.** As in Fig. S53 at FINL045 site.

**Photosynthesis parameters for FINL045**

[Figure]

[Figure]

[Figure]

[Figure]

[Figure]

[Figure]

**Figure S58.** As in Fig. S54 at FINL045 site.

[Figure]

**Figure S59.** Pearson correlation coefficients between tree growth observations and simulations at the Eastern Canadian taiga sites (Fig. 1) with MAIDEN using NRCAN (5') as climatic inputs (Table 2) for the 1950-2000 period with *QC_taiga* calibrated parameters from Gennaretti et al. (2017). Individual (left) and aggregated sites (right). The long-term decadal trends have been removed in observations and simulations. White inner circles stand for non-significant correlations (p-value > 0.05). Plain circles stand for significant correlations (p-value < 0.05).

[Figure]

**Figure S60.** Selected carbon allocation parameters value (Table S3) based on the calibration procedure detailed in Sect. 2.3.1 and 95% confidence interval of each parameter (computed based on all iterations of the third step of the calibration process, with a five iterations thinning and a burn-in period of 3000 iterations, see Sect. 2.3.1) for the five aggregated Eastern Canadian sites (Fig. 1b) and for all climatic datasets (Table 2) over the 1950-2000 time period. Dashed line corresponds to the parameter value at *QC_taiga* site from Gennaretti et al. (2017).

[Figure]

**Figure S61.** Selected photosynthesis parameters value (Table S3) based on the calibration procedure detailed in Sect. 2.3.1 and 95% confidence interval of each parameter (computed based on all iterations of the second step of the calibration process, with a five iterations thinning and a burn-in period of 1000 iterations, see Sect. 2.3.1) for the five aggregated Eastern Canadian sites (Fig. 1b) and for all climatic datasets (Table 2) over the 1950-2000 time period. Dashed line corresponds to the parameter value at *QC_taiga* site from Gennaretti et al. (2017).

[Figure]

[Figure]

[Figure]

**Figure S62.** WL42 (Fig. 1a). Ensemble spread of maximum temperature (Tmax sprd), minimum temperature (Tmin sprd) and precipitations (P sprd) for the NOAA-CIRES 20th Century Reanalysis V2c (Table 2) for the 1900-2000 time period.

[Figure]

**(a)** 1950-2000          **(b)** 1900-2000

**Figure S63.** Pearson correlation coefficients between tree growth observations and simulations at the Eastern Canadian taiga sites (Fig. 1a) with VS-Lite using the different climatic datasets described in Table 2 for the 1950-2000 (a) and 1900-2000 (b) calibration periods. White inner circles stand for non-significant correlations (p-value > 0.05).